# Fast TRAC:
# A Parameter-Free Optimizer for
# Lifelong Reinforcement Learning

**Aneesh Muppidi**
Harvard College
aneeshmuppidi@college.harvard.edu

**Zhiyu Zhang**
Harvard University
zhiyuz@seas.harvard.edu

**Heng Yang**
Harvard University
hankyang@seas.harvard.edu

## Abstract

A key challenge in lifelong reinforcement learning (RL) is the loss of plasticity, where previous learning progress hinders an agent's adaptation to new tasks. While regularization and resetting can help, they require precise hyperparameter selection at the outset and environment-dependent adjustments. Building on the principled theory of online convex optimization, we present a parameter-free optimizer for lifelong RL, called TRAC, which requires no tuning or prior knowledge about the distribution shifts. Extensive experiments on Procgen, Atari, and Gym Control environments show that TRAC works surprisingly well—mitigating loss of plasticity and rapidly adapting to challenging distribution shifts—despite the underlying optimization problem being nonconvex and nonstationary. Project website and code is available here.

## 1 Introduction

Spot, the agile robot dog, has been learning to walk confidently across soft, lush grass. But when Spot moves from the grassy field to a gravel surface, the small stones shift beneath her feet, causing her to stumble. When Spot tries to walk across a sandy beach or on ice, the challenges multiply, and her once-steady walk becomes erratic. Spot wants to adjust quickly to these new terrains, but the patterns she learned on grass are not suited to gravel, sand, or ice. Furthermore, she never knows when the terrain will change again and how different it will be, therefore must continually plan for the unknown while avoiding reliance on outdated experiences.

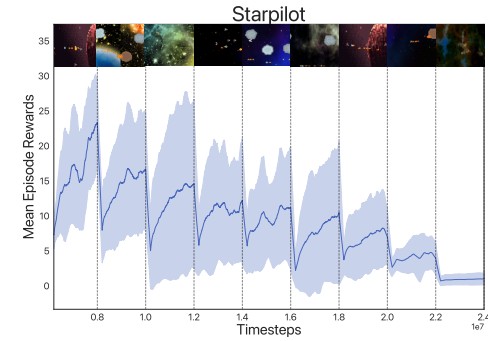

Figure 1: Severe loss of plasticity in Procgen (Starpilot). There is a steady decline in reward with each distribution shift.

Spot's struggle exemplifies a well-known and extensively studied challenge in real-world decision making: *lifelong reinforcement learning* (lifelong RL) Abel et al. (2024); Nath et al. (2023); Mendez et al. (2020); Xie & Finn (2022). In lifelong RL, the learning agent must continually acquire new knowledge to adapt to the nonstationarity of the environment. At first glance, there appears to be

38th Conference on Neural Information Processing Systems (NeurIPS 2024).

an obvious solution: given a policy gradient oracle, the agent could just keep running gradient descent nonstop. However, recent experiments have demonstrated an intriguing behavior called *loss of plasticity* (Dohare et al., 2021; Lyle et al., 2022; Abbas et al., 2023; Sokar et al., 2023): despite persistent gradient steps, such an agent can gradually lose its responsiveness to incoming observations. There are even extreme cases of loss of plasticity (known as *negative transfer* or *primacy bias*), where prior learning can significantly hamper the performance in new tasks (Nikishin et al., 2022; Ahn et al., 2024); see Figure 1 for an example. All these suggest that the problem is more involved than one might think.

From the optimization perspective, the above issues might be attributed to the *lack of stability* under gradient descent. That is, the weights of the agent's parameterized policy can drift far away from the origin (or a good initialization), leading to a variety of undesirable behaviors.[1] Fitting this narrative, it has been shown that simply adding a $L_2$ regularizer to the optimization objective (Kumar et al., 2023) or periodically resetting the weights (Dohare et al., 2021; Asadi et al., 2024; Sokar et al., 2023; Ahn et al., 2024) can help mitigate the problem. However, a particularly important limitation is their use of *hyperparameters*, such as the magnitude of the regularizer and the resetting frequency.[2] Good performance hinges on the suitable environment-dependent hyperparameter, but how can one confidently choose that *before* interacting with the environment? The classical cross-validation approach would violate the one-shot nature of lifelong RL (and online learning in general; see Chapter 1 of Orabona, 2023), since it is impossible to experience the same environment multiple times. This leads to the contributions of the present work.

**Contribution**  The present work addresses the key challenges in lifelong RL using the principled theory of *Online Convex Optimization* (OCO). Specifically, our contributions are two fold.

- **Algorithm: TRAC**  Building on a series of results in OCO (Cutkosky & Orabona, 2018; Cutkosky, 2019; Cutkosky et al., 2023; Zhang et al., 2024b), we propose a (hyper)-*parameter-free* optimizer for lifelong RL, called **TRAC** (Adap**T**ive **R**egulariz**A**tion in **C**ontinual environments). Intuitively, the idea is a refinement of regularization: instead of manually selecting the magnitude of regularization beforehand, TRAC chooses that in an online, data-dependent manner. From the perspective of OCO theory, TRAC is insensitive to its own hyperparameter, which means that no hyperparameter tuning is necessary in practice. Furthermore, as an optimization approach to lifelong RL, TRAC is compatible with any policy parameterization method.

- **Experiment**  Using *Proximal Policy Optimization* (PPO) (Schulman et al., 2017), we conduct comprehensive experiments on the instantiation of TRAC called TRAC PPO. A diverse range of lifelong RL environments are tested (based on Procgen, Atari, and Gym Control), with considerably larger scale than prior works. In settings where existing approaches (Abbas et al., 2023; Kumar et al., 2023; Nath et al., 2023) struggle, we find that TRAC PPO
  - mitigates mild and extreme loss of plasticity;
  - and rapidly adapts to new tasks when distribution shifts are introduced.

  Such findings might be surprising: the theoretical advantage of TRAC is motivated by the convexity in OCO, but lifelong RL is *both nonconvex and nonstationary* in terms of optimization.

**Organization**  Section 2 surveys the basics of lifelong RL. Section 3 introduces our parameter-free algorithm TRAC, and experiments are presented in Section 4. We defer the discussion of related works and results to Section 5. Finally, Section 6 concludes the paper.

## 2   Lifelong RL

As a sequential decision making framework, *reinforcement learning* (RL) is commonly framed as a *Markov Decision Process* (MDP) defined by the state space $\mathcal{S}$, the action space $\mathcal{A}$, the transition dynamics $P(s_{t+1}|s_t, a_t)$, and the reward function $R(s_t, a_t, s_{t+1})$. In the $t$-th round, starting from a state $s_t \in \mathcal{S}$, the learning agent needs to choose an action $a_t \in \mathcal{A}$ without knowing $P$ and $R$. Then, the environment samples a new state $s_{t+1} \sim P(\cdot|s_t, a_t)$, and the agent receives a *reward*

---

[1]Such as the inactivation of many neurons, due to the ReLU activation function (Abbas et al., 2023; Sokar et al., 2023).

[2]Indeed, hyperparameter selection, in general, is a well-known problem in lifelong as well as continual learning settings (De Lange et al., 2021).

$r_t = R(s_t, a_t, s_{t+1})$. There are standard MDP objectives driven by theoretical tractability, but from a practical perspective, we measure the agent's performance by its cumulative reward $\sum_{t=1}^{T} r_t$.

The standard setting above concerns a *stationary* MDP. Motivated by the prevalence of distribution shifts in practice, the present work studies a nonstationary variant called *lifelong* RL, where the transition dynamics $P_t$ and the reward function $R_t$ can vary over time. Certainly, one should not expect any meaningful "learning" against *arbitrary* unstructured nonstationarity. Therefore, we implicitly assume $P_t$ and $R_t$ to be *piecewise constant* over time, and each piece is called a *task* – just like our example of Spot in the introduction. The main challenge here is to transfer previous learning progress to new tasks. This is reasonable when tasks are similar, but we also want to reduce the degradation when tasks turn out to be very different.

**Lifelong RL as online optimization**  Deep RL approaches, including PPO (Schulman et al., 2017) and others, crucially utilize the idea of *policy parameterization*. Specifically, a policy refers to the distribution of the agent's action $a_t$ (conditioned on the historical observations), and we use $\theta_t \in \mathbb{R}^d$ to denote the parameterizing *weight vector*. After sampling $a_t$ and receiving new observations, the agent could define a *loss function* $J_t(\theta)$ that characterizes the "hypothetical performance" of each weight $\theta \in \mathbb{R}^d$. Then, by computing the *policy gradient* $g_t = \nabla J_t(\theta_t)$, one could apply a *first order optimization algorithm*[3] OPT to obtain the updated weight, $\theta_{t+1} = \text{OPT}(\theta_t, g_t)$.

For the rest of this paper, we will work with such an abstraction. The feedback of the environment is treated as a *policy gradient oracle* $\mathcal{G}$, which maps the time $t$ and the current weight $\theta_t$ into a policy gradient $g_t = \mathcal{G}(t, \theta_t)$. Our goal is to design an optimizer OPT well suited for lifelong RL.

**Lifelong vs. Continual**  In the RL literature, the use of "lifelong" and "continual" varies significantly across studies, which may lead to confusion. Abel et al. (2024) characterized *continual reinforcement learning* (CRL) as a never-ending learning process. However, much of the literature cited under CRL, such as (Abbas et al., 2023; Ahn et al., 2024), primarily focuses on the problem of *backward transfer* (avoiding catastrophic forgetting). Various policy-based architectures, such as those proposed by Rolnick et al. (2019); Schwarz et al. (2018); Nath et al. (2023), focus on tackling this issue. Conversely, the present work addresses the problem of *forward transfer*, which refers to the rapid adaptation to new tasks. Because of this we use "lifelong" rather than "continual" in our exposition, similar to (Thrun, 1996; Abel et al., 2018b; Julian et al., 2020).

## 3    Method

Inspired by (Cutkosky et al., 2023), we study lifelong RL by exploiting its connection to *Online Convex Optimization* (OCO; Zinkevich, 2003). The latter is a classical theoretical problem in online learning, and much effort has been devoted to designing *parameter-free* algorithms that require minimum tuning or prior knowledge (Streeter & Mcmahan, 2012; McMahan & Orabona, 2014; Orabona & Pál, 2016; Foster et al., 2017; Cutkosky & Orabona, 2018; Mhammedi & Koolen, 2020; Chen et al., 2021; Jacobsen & Cutkosky, 2022). The surprising observation of Cutkosky et al. (2023) is that several algorithmic ideas closely tied to the convexity of OCO can actually improve the nonconvex deep learning training, suggesting certain notions of "near convexity" on its

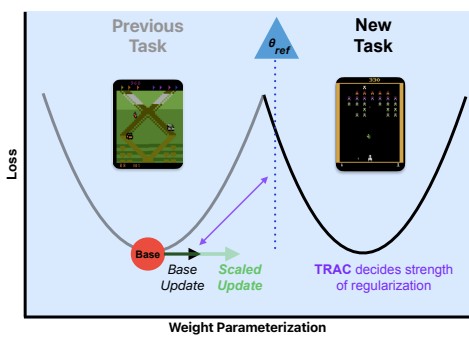

Figure 2: Visualization of TRAC's key idea.

loss landscape. We find that lifelong RL (which is *both nonconvex and nonstationary* in terms of optimization) exhibits a similar behavior, therefore a particularly strong algorithm (named TRAC) can be obtained from principled results in parameter-free OCO. Let us start from the background.

**Basics of (parameter-free) OCO**  As a standalone theoretical topic, OCO concerns a sequential optimization problem where the convex loss function $l_t$ can vary arbitrarily over time. In the $t$-th

---

[3]Formally, a dynamical system that given its state $\theta_t$ and input $g_t$ outputs the new state $\text{OPT}(\theta_t, g_t)$.

iteration, the optimization algorithm picks an iterate $x_t$ and then observes a gradient $g_t = \nabla l_t(x_t)$. Motivated by the pursuit of "convergence" in optimization, the standard objective is to guarantee low (i.e., sublinear in $T$) *static regret*, defined as

$$\text{Regret}_T(l_{1:T}, u) := \sum_{t=1}^{T} l_t(x_t) - \sum_{t=1}^{T} l_t(u),$$

where $T$ is the total number of rounds, and $u$ is a *comparator* that the algorithm does not know beforehand. In other words, the goal is to make $\text{Regret}_T(l_{1:T}, u)$ small for *all* possible loss sequence $l_{1:T}$ and comparator $u$. Note that for *nonstationary* OCO problems analogous to lifelong RL, it is better to consider a different objective called the *discounted regret*. Algorithms there mostly follow the same principle as in the stationary setting, just wrapped by *loss rescaling* (Zhang et al., 2024a).

For minimizing static regret, classical *minimax* algorithms like gradient descent (Zinkevich, 2003) would assume a small *uncertainty set* $\mathcal{U}$ at the beginning. Then, by setting the hyperparameter (such as the learning rate) according to $\mathcal{U}$, it is possible to guarantee sublinear *worst case regret*,

$$\max_{(l_{1:T}, u) \in \mathcal{U}} \text{Regret}_T(l_{1:T}, u) = o(T). \tag{1}$$

In contrast, parameter-free algorithms use very different strategies[4] to bound $\text{Regret}_T(l_{1:T}, u)$ directly (without taking the maximum) by a function of both $l_{1:T}$ and $u$. The resulting bound is more refined than Eq.(1) (Orabona, 2023, Chapter 9), and crucially, since there is no need to pick an uncertainty set $\mathcal{U}$, much less hyperparameter tuning is needed. This is where its name comes from.

**TRAC for Lifelong RL:** In lifelong RL, a key issue is the excessive drifting of weights $\theta_t$, which can detrimentally affect adapting to new tasks. To address this, TRAC enforces proximity to a well-chosen reference point $\theta_{\text{ref}}$, providing a principled solution derived from a decade of research in parameter-free OCO. Unlike traditional methods such as $L_2$ regularization or resetting, TRAC avoids hyperparameter tuning, utilizing the properties of OCO to maintain weight stability and manage the drift effectively.

The core of TRAC, similar to other parameter-free optimizers, incorporates three techniques:

- **Direction-Magnitude Decomposition**: Inspired by Cutkosky & Orabona (2018), this technique employs a carefully designed one-dimensional algorithm, the "parameter-free tuner," atop a base optimizer. This setup acts as a data-dependent regularizer, controlling the extent to which the iterates deviate from their initialization, thereby minimizing loss of plasticity, which is crucial given the high plasticity at the initial policy parameterization (Abbas et al., 2023).

- **Erfi Potential Function**: Building on the previous concept, the tuner utilizes the Erfi potential function, as developed by Zhang et al. (2024a). This function is crafted to effectively balance the distance of the iterates from both the origin and the empirical optimum. It manages the update magnitude by focusing on the gradient projection along the direction $\theta_t - \theta_{\text{ref}}$.

- **Additive Aggregation:** The tuner above necessitates discounting. Thus, we employ Additive Aggregation by Cutkosky (2019). This approach enables the combination of multiple parameter-free OCO algorithms, each with different discount factors, to approximate the performance of the best-performing algorithm. Importantly, it facilitates the automatic selection of the optimal discount factor during training.

These three components crucially work together to guarantee good regret bounds in the convex setting and are the minimum requirement for any reasonable parameter-free optimizer.

Without going deep into the theory, here is an overview of the important ideas (also see Figure 2 for a visualization).

- First, TRAC is a meta-algorithm that operates on top of a "default" optimizer BASE. It can simply be gradient descent with a constant learning rate, or ADAM (Kingma & Ba, 2014) as in our experiments. Applying BASE alone would be equivalent to enforcing the scaling parameter $S_{t+1} \equiv 1$ in TRAC, but this would suffer from the drifting of $\theta_{t+1}^{\text{Base}}$ (and thus, the weight $\theta_{t+1}$).

---

[4]The key difference with gradient descent is the use of intricate (non-$L_2$) regularizers. See (Fang et al., 2022; Jacobsen & Cutkosky, 2022) for a theoretical justification of their importance.

---

**Algorithm 1** TRAC: Parameter-free Adaption for Continual Environments.

1: **Input:** A policy gradient oracle $\mathcal{G}$; a first order optimization algorithm BASE; a reference point $\theta_{\mathrm{ref}} \in \mathbb{R}^d$; $n$ discount factors $\beta_1, \ldots, \beta_n \in (0, 1]$ (default: $0.9, 0.99, \ldots, 0.999999$).

2: **Initialize:** Create $n$ copies of Algorithm 2, denoted as $\mathcal{A}_1, \ldots, \mathcal{A}_n$. For each $j \in [1 : n]$, $\mathcal{A}_j$ uses the discount factor $\beta_j$. Initialize the algorithm BASE at $\theta_{\mathrm{ref}}$. Let $\theta_1 = \theta_{\mathrm{ref}}$.

3: **for** $t = 1, 2, \ldots$ **do**

4:     Obtain the $t$-th policy gradient $g_t = \mathcal{G}(t, \theta_t) \in \mathbb{R}^d$.

5:     Send $g_t$ to BASE as its $t$-th input, and get its output $\theta_{t+1}^{\mathrm{Base}} \in \mathbb{R}^d$.

6:     For all $j \in [1 : n]$, send $\langle g_t, \theta_t - \theta_{\mathrm{ref}} \rangle$ to $\mathcal{A}_j$ as its $t$-th input, and get its output $s_{t+1,j} \in \mathbb{R}$.

7:     Define the scaling parameter $S_{t+1} = \sum_{j=1}^{n} s_{t+1,j}$.

8:     Update the weight of the policy,

$$\theta_{t+1} = \theta_{\mathrm{ref}} + \left(\theta_{t+1}^{\mathrm{Base}} - \theta_{\mathrm{ref}}\right) S_{t+1}.$$

9: **end for**

---

**Algorithm 2** 1D Discounted Tuner of TRAC.

1: **Input:** Discount factor $\beta \in (0, 1]$; small value $\varepsilon > 0$ (default: $10^{-8}$).

2: **Initialize:** The running variance $v_0 = 0$; the running (negative) sum $\sigma_0 = 0$.

3: **for** $t = 1, 2, \ldots$ **do**

4:     Obtain the $t$-th input $h_t$.

5:     Let $v_t = \beta^2 v_{t-1} + h_t^2$, and $\sigma_t = \beta \sigma_{t-1} - h_t$.

6:     Select the $t$-th output

$$s_{t+1} = \frac{\varepsilon}{\mathrm{erfi}(1/\sqrt{2})} \mathrm{erfi}\left(\frac{\sigma_t}{\sqrt{2v_t} + \varepsilon}\right),$$

    where erfi is the *imaginary error function* queried from standard software packages.

7: **end for**

---

- To fix this issue, TRAC uses the tuner (Algorithm 2) to select the scaling parameter $S_{t+1}$, making it *data-dependent*. Typically $S_{t+1}$ is within $[0, 1]$ (see Figure 17 to 19), therefore essentially, we define the updated weight $\theta_{t+1}$ as a *convex combination* of the BASE's weight $\theta_t^{\mathrm{Base}}$ and the reference point $\theta_{\mathrm{ref}}$,

$$\theta_{t+1} = S_{t+1} \cdot \theta_{t+1}^{\mathrm{Base}} + (1 - S_{t+1})\theta_{\mathrm{ref}}.$$

  This brings the weight closer to $\theta_{\mathrm{ref}}$, which is known to be "safe" (i.e., not overfitting any particular lifelong RL task), although possibly conservative.

- To inject the right amount of conservatism without hyperparameter tuning, the tuner (Algorithm 2) applies an unusual decision rule based on the erfi function. Theoretically, this is known to be optimal in an idealized variant of OCO (Zhang et al., 2022, 2024b), but removing the idealized assumptions requires a tiny bit of extra conservatism, which is challenging (and not necessarily practical). Focusing on the lifelong RL problem that considerably deviates from OCO, we simply apply the erfi decision rule as is. This is loosely motivated by deep learning training dynamics, e.g., (Cohen et al., 2020; Ahn et al., 2023; Andriushchenko et al., 2023), where an aggressive optimizer is often observed to be better.

- Finally, the tuner requires a discount factor $\beta$. This crucially controls the strength of regularization (elaborated next), but also introduces a hyperparameter tuning problem. Following (Cutkosky, 2019), we aggregate tuners with different $\beta$ (on a log-scaled grid) by simply summing up their outputs. This is justified by the *adaptivity* of the tuner itself: in OCO, if we add a parameter-free algorithm $\mathcal{A}_1$ to any other algorithm $\mathcal{A}_2$ that already works well, then $\mathcal{A}_1$ can automatically identify this and "tune down" its aggressiveness, such that $\mathcal{A}_1 + \mathcal{A}_2$ still performs as well as $\mathcal{A}_2$.

**Connection to regularization** Despite its nested structure, TRAC can actually be seen as a parameter-free refinement of $L_2$ regularization (Kumar et al., 2023). To concretely explain this intuition, let us consider the following two optimization dynamics.

- First, suppose we run gradient descent with learning rate $\eta$, on the policy gradient sequence $\{g_t\}$ with the $L_2$ regularizer $\frac{\lambda}{2}\|\theta - \theta_{\text{ref}}\|^2$. Quantitatively, it means that starting from the $t$-th weight $\theta_t$,

$$\theta_{t+1} = \theta_t - \eta\left[g_t + \lambda\left(\theta_t - \theta_{\text{ref}}\right)\right], \quad \implies \quad \theta_{t+1} - \theta_{\text{ref}} = (1 - \lambda\eta)\left(\theta_t - \theta_{\text{ref}}\right) - \eta g_t. \quad (2)$$

  That is, the updated weight $\theta_{t+1}$ is determined by a $(1 - \lambda\eta)$-discounting with respect to the reference point $\theta_{\text{ref}}$, followed by a gradient step $-\eta g_t$.

- Alternatively, consider applying the following simplification of TRAC on the same policy gradient sequence $\{g_t\}$: $(i)$ BASE is still gradient descent with learning rate $\eta$; $(ii)$ there is just one discount factor $\beta$; and $(iii)$ the one-dimensional tuner (Algorithm 2) is replaced by the $\beta$-discounted gradient descent with learning rate $\alpha$, i.e., $S_{t+1} = \beta S_t - \alpha h_t$. In this case, we have

$$\begin{aligned}
\theta_{t+1} - \theta_{\text{ref}} &= S_{t+1}\left(\theta_{t+1}^{\text{Base}} - \theta_{\text{ref}}\right) \\
&= (\beta S_t - \alpha h_t)\left(\theta_t^{\text{Base}} - \theta_{\text{ref}} - \eta g_t\right) \\
&= \left(\beta - \alpha S_t^{-1} h_t\right)\left(\theta_t - \theta_{\text{ref}}\right) - \eta S_{t+1} g_t. \qquad \text{(mildly assuming } S_t \neq 0\text{)}
\end{aligned}$$

  Notice that $S_t$ is a $\beta$-discounted sum of $\alpha h_1, \dots, \alpha h_{t-1}$, thus in the typical situation of $\beta \approx 1$ one might expect $\alpha h_t \ll |S_t|$. Then, the resulting update of $\theta_{t+1}$ is similar to Eq.(2), with quantitative changes on the "effective discounting" $1 - \lambda\eta \to \beta$, and the "effective learning rate" $\eta \to \eta S_{t+1}$.

The main message here is that under a simplified setting, TRAC is almost equivalent to $L_2$ regularization. The latter requires choosing the hyperparameters $\lambda$ and $\eta$, and similarly, the above *simplified* TRAC requires choosing $\beta$ and $\eta$. Going beyond this simplification, the actual TRAC removes the tuning of $\beta$ using aggregation, and the tuning of $\eta$ using the erfi decision rule.

**On the hyperparameters** Although TRAC is called "parameter-free", it still needs the $\beta$-grid, the constant $\varepsilon$ and the algorithm BASE as inputs. The idea is that TRAC is particularly insensitive to such choices, as supported by the OCO theory. As the result, the generic default values recommended by Cutkosky et al. (2023) are sufficient in practice. We note that those are proposed for training supervised deep learning models, thus should be agnostic to the lifelong RL applications we consider.

## 4 Experiment

Does TRAC experience the common pitfalls of loss of plasticity? Does it rapidly adapt to distribution shifts? To answer these questions, we test TRAC in empirical RL benchmarks such as vision-based games and physics-based control environments in lifelong settings (Figure 3). Specifically, we instantiate PPO with two different optimizers: ADAM with constant learning rate for baseline comparison, and TRAC for our proposed method (with exactly the same ADAM as the input BASE). We also test ADAM PPO with *concatenated ReLU activations* (CReLU; Shang et al., 2016), previously shown to mitigate loss of plasticity in certain deep RL settings (Abbas et al., 2023). Our numerical results are summarized in Table 1. Across every lifelong RL setting, we observe substantial improvements in the cumulative episode reward by using TRAC PPO compared to ADAM PPO or CReLU. Below are the details, with more in the Appendix.

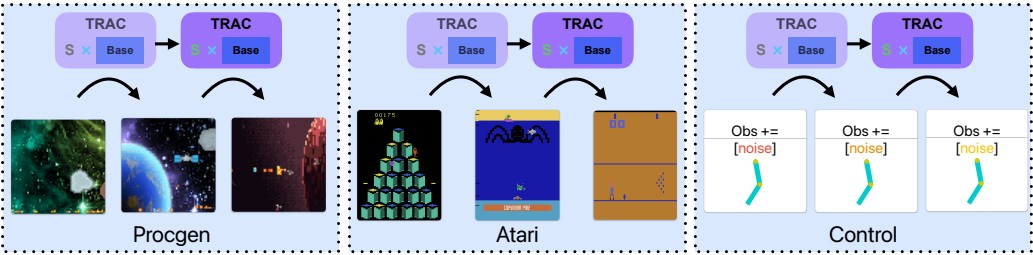

Figure 3: Experimental setup for lifelong RL.

**Procgen** We first evaluate on OpenAI Procgen, a suite of 16 procedurally generated game environments (Cobbe et al., 2020). We introduce distribution shifts by sampling a new procedurally generated level of the current game every 2 million time steps, treating each level as a distinct task.

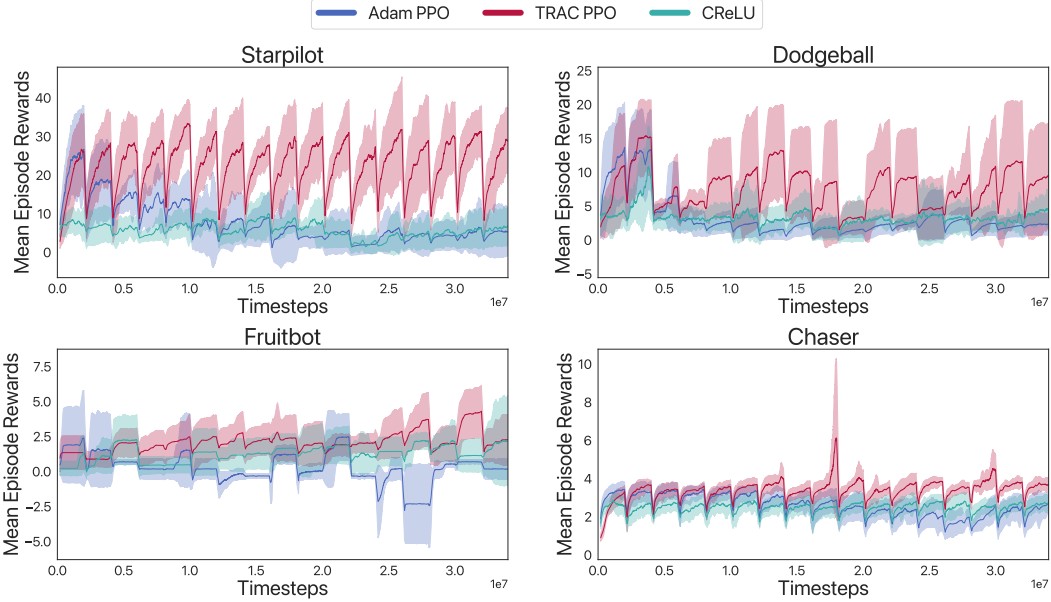

Figure 4: Reward in the lifelong Procgen environments for StarPilot, Dodgeball, Fruitbot, and Chaser. There is a steady loss of plasticity in agents using ADAM PPO and CReLU, characterized by their inability to maintain performance through succesive Procgen levels. In contrast, TRAC avoids this loss of plasticity, quickly achieving high performance with each new task.

We evaluate game environments including StarPilot, Dodgeball, Fruitbot, and Chaser. In all of these environments, we observe in Figure 4 that both ADAM PPO and CReLU encounter a continually degrading loss of plasticity as these distribution shifts are introduced. In contrast, TRAC PPO avoids this loss of plasticity, which contributes to its rapid reward increase when adapting to new levels. In the cumulative reward across all the Procgen levels, TRAC PPO reveals normalized average improvements of 3,212.42% and 120.88% over ADAM PPO and CReLU respectively (see Table 1). For later levels, in all games, TRAC PPO's reward does not decline as sharply as the baselines, potentially indicating positive transfer of skills from one level to the next.

One key advantage of TRAC is that it functions as an optimizer, making it orthogonal to various policy methods such as PPO, as well as other baselines like Online EWC (Schwarz et al., 2018), IMPALA (Espeholt et al., 2018), Modulating Masks (Nath et al., 2023), and CLEAR (Rolnick et al., 2019). In Appendix C, we evaluate these methods using both TRAC and ADAM on the Procgen setup. We find that in every environment, TRAC improves the performance of these algorithms.

**Atari**   The Arcade Learning Environment (ALE) Atari 2600 benchmark is a collection of classic arcade games designed to assess reinforcement learning agents' performance across a range of diverse gaming scenarios (Bellemare et al., 2013). We introduce distribution shifts by switching to a new Atari game every 4 million timesteps, where each game switch introduces a new task. This benchmark is more challenging compared to OpenAI Procgen: it requires the agent to handle distribution shifts in both the input (state) and the target (reward).

In this experiment, we assessed two online settings distinguished by games with action spaces of 6 and 9. From Figure 5, both ADAM PPO and CReLU sometimes failed to learn in certain games. In contrast, TRAC PPO shows a substantial increase in reward over different games compared to the baselines. For example, during the first 12 million steps (3 games) in Atari 6, TRAC PPO not only achieves a significantly higher mean reward but also demonstrates rapid reward increase. Over both experiment settings, TRAC PPO shows an average normalized improvement of 329.73% over ADAM PPO and 68.71% over CReLU (Table 1). In rare instances, such as the last 2 million steps of Atari 6, CReLU performs comparably to TRAC PPO. This observation aligns with findings from (Abbas et al., 2023), which noted that CReLU tends to avoid plasticity loss in continual Atari setups.

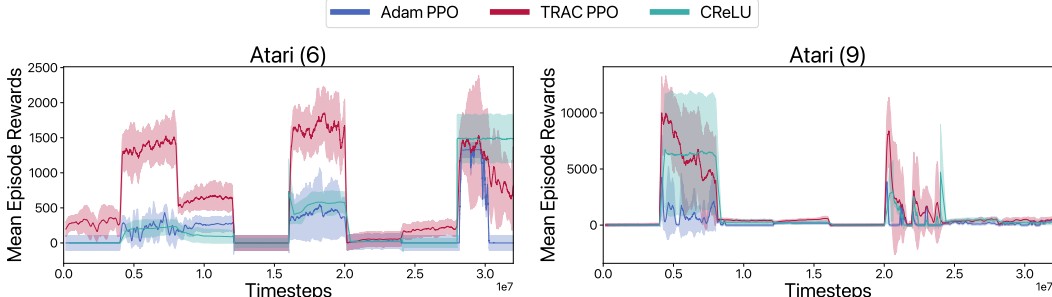

Figure 5: Reward in the lifelong Atari environments, across games with action spaces of 6 and 9. TRAC PPO rapidly adapts to new tasks, in contrast to the ADAM PPO and CReLU which struggle to achieve high reward, indicating mild loss of plasticity.

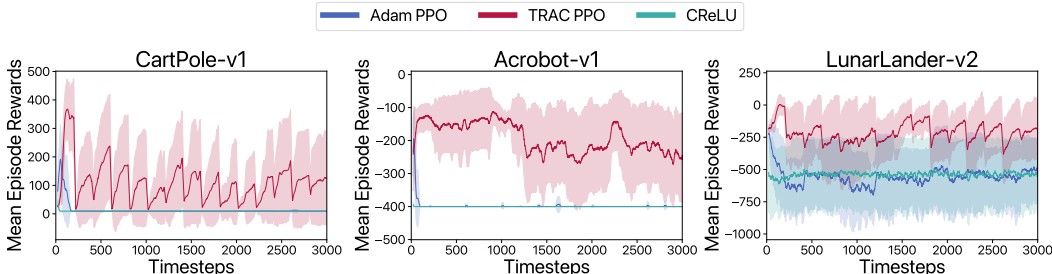

Figure 6: Reward performance across CartPole, Acrobot, and LunarLander Gym Control tasks. Both ADAM PPO and CReLU experience extreme plasticity loss, failing to recover after the initial distribution shift. Conversely, TRAC PPO successfully avoids such plasticity loss, rapidly adapting when facing extreme distribution shifts.

**Gym Control** We use the CartPole-v1 and Acrobot-v1 environments from the Gym Classic Control suite, along with LunarLander-v2 from Box2d Control. To introduce distribution shifts, Mendez et al. (2020) periodically alters the environment dynamics. Although such distribution shifts pose only mild challenges for robust methods like PPO with ADAM (Appendix D). We instead implement a more challenging form of distribution shift. Every 200 steps we perturb each observation dimension with random noise within a range of $\pm 2$, treating each perturbation phase as a distinct task.

Table 1: Cumulative sum of mean episode reward for TRAC PPO, ADAM PPO, and CReLU on Procgen, Atari, and Gym Control environments. Rewards are scaled by $10^5$; higher is better.

| Environment | ADAM PPO | CReLU | TRAC PPO (Ours) |
|---|---|---|---|
| Starpilot | 3.4 | 3.6 | **12.5** |
| Dodgeball | 1.9 | 2.3 | **5.2** |
| Chaser | 1.4 | 1.7 | **2.2** |
| Fruitbot | 0.1 | 1.0 | **1.8** |
| CartPole | 5.1 | 1.2 | **39.6** |
| Acrobot | −14.3 | −13.9 | **−12.9** |
| LunarLander | −21.7 | −19.4 | **−8.6** |
| Atari 6 | 3.1 | 4.8 | **10.5** |
| Atari 9 | 3.9 | 17.0 | **20.2** |

Here (Figure 6), we notice a peculiar behavior after introducing the first distribution shift in both ADAM PPO and CReLU: policy collapse. We describe this as an *extreme* form of loss of plasticity. Surprisingly, TRAC PPO remains resistant to these extreme distribution shifts. As we see in the Acrobot experiment, TRAC PPO shows minimal to no policy damage after the first few distribution shifts, whereas ADAM PPO and CReLU are unable to recover a policy at all. We investigate if TRAC's behavior here indicates positive transfer in Appendix A. Across the three Gym Control environments,

TRAC PPO shows an average normalized improvement of 204.18% over ADAM PPO and 1044.24% over CReLU (Table 1).

## 5  Discussion

**Related work**   Combating loss of plasticity has been studied extensively in lifelong RL. A typical challenge for existing solutions is the tuning of their hyperparameters, which requires prior knowledge on the nature of the distribution shift, e.g., (Asadi et al., 2024; Nath et al., 2023; Nikishin et al., 2024; Sokar et al., 2023; Mesbahi et al., 2024). An architectural modification called CReLU is studied in (Abbas et al., 2023), but our experiments suggest that its benefit might be specific to the Atari setup. Besides, Abel et al. (2018a,b) presented a theoretical analysis of skill transfer in lifelong RL, based on value iteration. Moreover, related contributions in nonstationary RL, where reward and state transition functions also change unpredictably, are limited to theoretical sequential decision-making settings with a focus on establishing complexity bounds (Roy et al., 2019; Cheung et al., 2020; Wei & Luo, 2021; Mao et al., 2020).

Our algorithm TRAC builds on a long line of works on parameter-free OCO (see Section 3). To our knowledge, the only existing work applying parameter-free OCO to RL is (Jacobsen & Chan, 2021), which focuses on estimating the value function (i.e., policy evaluation). Our scope is different, focusing on empirical RL in lifelong problems by exploring the key connection between parameter-free OCO and regularization.

Particularly, we are inspired by the MECHANIC algorithm from (Cutkosky et al., 2023), which goes beyond the traditional convex setting of parameter-free OCO to handle stationary deep learning optimization tasks. Lifelong reinforcement learning, however, introduces a layer of complexity with its inherent nonstationarity. Furthermore, compared to MECHANIC, TRAC improves the scale tuner there (which is based on the *coin-betting* framework; Orabona & Pál, 2016) by the erfi algorithm that enjoys a better OCO performance guarantee. As an ablation study, we empirically compare TRAC and MECHANIC in the Appendix G (Table 3). We find that TRAC is slightly better, but both algorithms can mitigate the loss of plasticity, suggesting the effectiveness of the general "parameter-free" principle in lifelong RL.

**TRAC encourages positive transfer**   In our experiments, we observe that TRAC's reward decline due to distribution shifts is less severe than that of baseline methods. These results may suggest TRAC facilitates positive transfer between related tasks. To investigate this further, we compared TRAC to a privileged weight-resetting approach, where the network's parameters are reset for each new task, in the Gym Control environments (see Appendix A). Our results show that TRAC maintains higher rewards during tasks than privileged weight-resetting and avoids declining to the same low reward levels as privileged weight-resetting at the start of a new task (Figure 8).

**On the choice of $\theta_{\text{ref}}$**   In general, the reference point $\theta_{\text{ref}}$ should be good or "safe" for TRAC to perform effectively. One might presume that achieving this requires "warmstarting", or pre-training using the underlying BASE optimizer. While our experiments validate that such warmstarting is indeed beneficial (Appendix B), our main experiments show that even a random initialization of the policy's weight serves as a *good enough* $\theta_{\text{ref}}$, even when tasks are similar (Figure 4).

This observation aligns with discussions by Lyle et al. (2023), Sokar et al. (2023), and Abbas et al. (2023), who suggested that persistent gradient steps away from a random initialization can deactivate ReLU activations, leading to activation collapse and loss of plasticity in neural networks. Our results also support Kumar et al. (2023)'s argument that maintaining some weights close to their initial values not only prevents dead ReLU units but also allows quick adaptation to new distribution shifts.

**Tuning $L_2$ regularization**   The success of TRAC suggests that an adaptive form of regularization—anchoring to the reference point $\theta_{\text{ref}}$—may suffice to counteract both mild and extreme forms of loss of plasticity. From this angle, we further elaborate the limitation of the $L_2$ regularization approach considered in (Kumar et al., 2023). It requires selecting a regularization strength parameter $\lambda$ through cross-validation, which is incompatible with the one-shot nature of lifelong learning settings. Furthermore, it is nontrivial to select the search grid: for example, we tried the $\lambda$-grid suggested by (Kumar et al., 2023), and there is no effective $\lambda$ value within the grid for the lifelong RL environments we consider. All the values are too small.

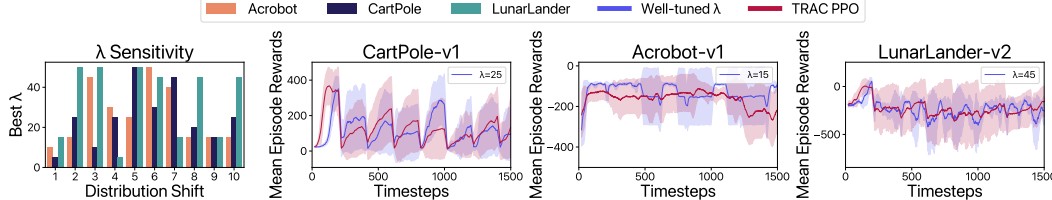

Figure 7: For each Gym Control environment and the initial ten tasks, we identified the best $\lambda$, which is the regularization strength that maximizes reward for each task's specific distribution shift. We also determined the best overall (well-tuned) $\lambda$ for each environment. The results demonstrate that each environment and each task's distribution shift is sensitive to different $\lambda$ and that TRAC PPO performs competitively with each environment's well-tuned $\lambda$.

Continuing this reasoning, we conduct a hyperparameter search for $\lambda$, over various larger values $[0.2, 0.8, 1, 5, 10, 15, 20, 25, 30, 35, 40, 45, 50]$. Given the expense of such experiments, only the more sample-efficient control environments are considered. We discover that each environment and task responds uniquely to these regularization strengths (see bar plot of $\lambda$ values in Figure 7). This highlights the challenges of tuning $\lambda$ in a lifelong learning context, where adjusting for each environment, let alone each distribution shift, would require extensive pre-experimental analysis.

In contrast, TRAC offers a parameter-free solution that adapts dynamically with the data in an online manner. The scaling output of TRAC adjusts autonomously to the ongoing conditions, consistently competing with well-tuned $\lambda$ values in the various environments, as demonstrated in the reward plots for CartPole, Acrobot, and LunarLander (Figure 7).

**TRAC compared to other plasticity methods**    Both layer normalization and plasticity injection Nikishin et al. (2024); Lyle et al. (2023) have been shown to combat plasticity loss. For instance, Appendix E Figure 15 demonstrates that both layer normalization and plasticity injection are effective at reducing plasticity loss when applied to the CartPole environment using ADAM as a baseline optimizer. We implemented plasticity injection following the methodology laid out by Nikishin et al. (2024), where plasticity is injected at the start of every distribution shift. While this approach does help in reducing the decline in performance due to plasticity loss, our results indicate that it is consistently outperformed by TRAC across all three control environments—CartPole, Acrobot, and LunarLander. Moreover, while layer normalization improves ADAM's performance, it too is outperformed by TRAC across the same control settings (Figure 15). Notably, combining layer normalization with TRAC resulted in the best performance gains.

**Near convexity of lifelong RL**    Our results demonstrate the rapid adaptation of TRAC, in lifelong RL problems with complicated function approximation. From the perspective of optimization, the latter requires tackling both nonconvexity and nonstationarity, which is typically regarded intractable in theory. Perhaps surprisingly, when approaching this complex problem using the theoretical insights from OCO, we observe compelling results. This suggests a certain "hidden convexity" in this problem, which could be an exciting direction for both theoretical and empirical research (e.g., policy gradient methods provably converge to global optimizers in linear quadratic control (Hu et al., 2023)).

**Limitations**    While TRAC offers robust adaptability in nonstationary environments, it can exhibit suboptimal performance at the outset. In the early stages of deployment, TRAC might underperform compared to the baseline optimizer. We address this by proposing a warmstarting solution detailed in Appendix B, which helps increase the initial performance gap.

## 6   Conclusion

In this work, we introduced TRAC, a parameter-free optimizer for lifelong RL that leverages the principles of OCO. Our approach dynamically refines regularization in a data-dependent manner, eliminating the need for hyperparameter tuning. Through extensive experimentation in Procgen, Atari, and Gym Control environments, we demonstrated that TRAC effectively mitigates loss of plasticity and rapidly adapts to new distribution shifts, where baseline methods fail. TRAC's success leads to a compelling takeaway: empirical lifelong RL scenarios may exhibit more convex properties than previously appreciated, and might inherently benefit from parameter-free OCO approaches.

# 7  Acknowledgments

We thank Ashok Cutkosky for insightful discussions on online optimization in nonstationary settings. We are grateful to David Abel for his thoughtful insights on loss of plasticity in relation to lifelong reinforcement learning. We appreciate Kaiqing Zhang and Yang Hu for their comments on theoretical and nonstationary RL. This project is partially funded by Harvard University Dean's Competitive Fund for Promising Scholarship.

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

# Appendix

## A   TRAC Encourages Positive Transfer

To explore whether TRAC encourages positive transfer, we introduce a privileged weight-reset baseline. This baseline is "privileged" in the sense that it knows when a distribution shift is introduced and resets the parameters to a random initialization at the start of each new task. We applied this baseline to three Gym control tasks: CartPole-v1, Acrobot-v1, and LunarLander-v2, and compared it to TRAC PPO and ADAM PPO, as shown in Figure 8.

We observe that the privileged weight-reset baseline exhibits spikes in reward at the beginning of each new task. Surprisingly, TRAC maintains even higher rewards than the privileged weight-reset baseline, even at its peak learning phases. Additionally, TRAC's reward does not decline to the reward seen at the start of new tasks with privileged weight-resetting (TRAC does not have to "start over" with each task), suggesting that TRAC successfully transfers skills positively between tasks.

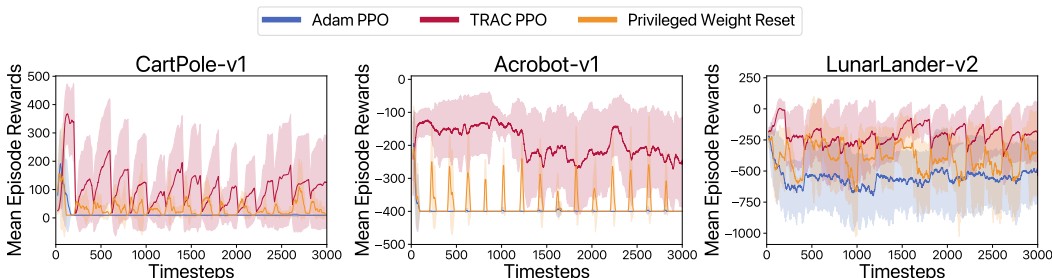

Figure 8: Reward comparison of TRAC PPO, ADAM PPO, and privileged weight-resetting on Cartpole-v1, Acrobot-v1, and LunarLander-v2. TRAC PPO encourages positive transfer between tasks.

## B   Warmstarting

In our theoretical framework, we hypothesize that a robust parameter initialization, denoted as $\theta_{\text{ref}}$, could enhance the performance of our models, suggesting that empirical implementations might benefit from initializing parameters using a base optimizer such as ADAM prior to deploying TRAC. Contrary to this assumption, our experimental results detailed in Section 4 reveal that warmstarting is not essential for TRAC's success. Below, we examine the performance of ADAM PPO and TRAC PPO when warmstarted.

Both TRAC PPO and ADAM PPO were warmstarted using ADAM for the initial 150,000 steps in all games for the Atari and Procgen environments, and for the first 30 steps in the Gym Control experiments. As seen in Figure 9, in games like Starpilot, Fruitbot, and Dodgeball, TRAC PPO surpasses ADAM PPO in the first level/task of the online setup, with its performance closely matching that of ADAM PPO in Chaser. Importantly, TRAC PPO continues to avoid the loss of plasticity encountered by ADAM PPO, even when both are warmstarted. This makes sense since all of the distributions share some foundational game dynamics; the initial learning phases likely explore these dynamics, so leveraging a good parameter initialization to regularize in this early region can be beneficial for TRAC—we observe that forward transfer occurs somewhat in later level distribution shifts as the reward does not drop back to zero where it initially started from.

Our findings indicate that warmstarting does not confer a significant advantage in the Atari games. This makes sense because a parameter initialization that is good in one game setting is likely a random parameterization for another setting, which is equivalent to the setup without warmstarting where TRAC regularizes towards a random parameter initialization. In the Gym Control experiments although warmstarted TRAC PPO manages to avoid the extreme plasticity loss and policy collapse seen in warmstarted ADAM PPO, it does not perform as well as non-warmstarted TRAC PPO. This result underscores that the efficacy of warmstarting is environment-specific and highlights the

challenge in predicting when ADAM PPO may achieve a parameter initialization that is advantageous for TRAC PPO to regularize towards.

From an overall perspective, warmstarting TRAC PPO in every setting still shows substantial improvement over ADAM PPO (Table 2).

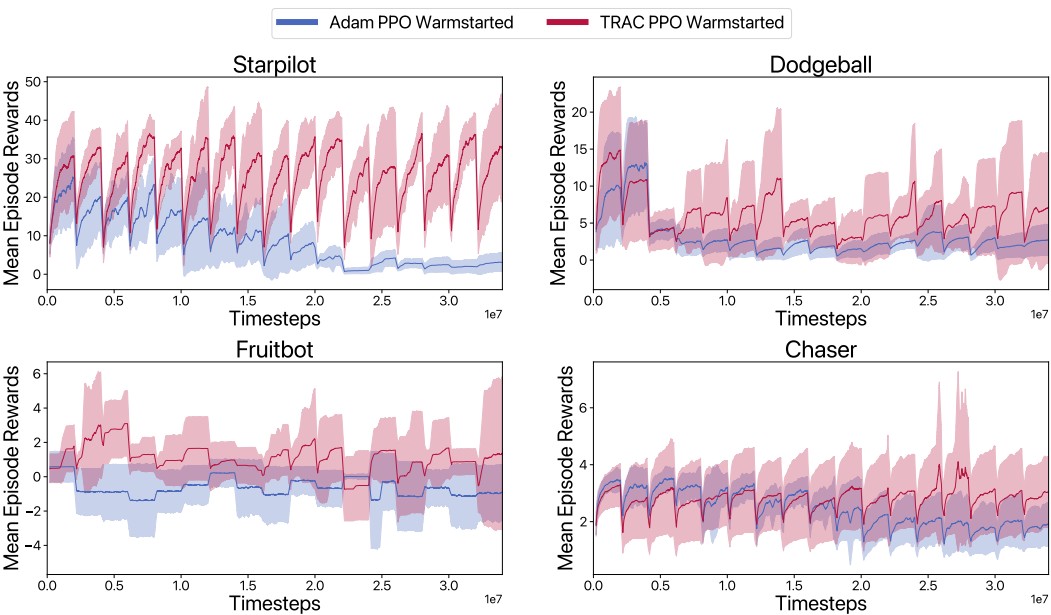

Figure 9: Reward in the lifelong Procgen environments for StarPilot, Dodgeball, Fruitbot, and Chaser with warmstarted TRAC PPO and warmstarted ADAM PPO. Inital performance of TRAC PPO is improved with warmstarting and continues to avoid loss of plasticity.

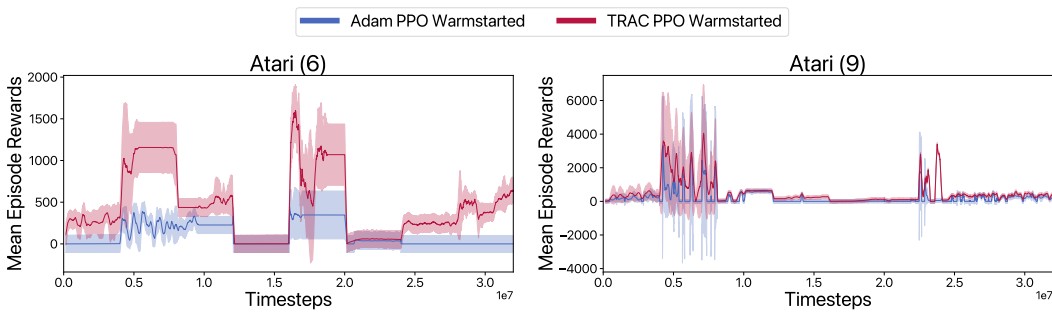

Figure 10: Reward in the lifelong Atari environments with warmstarted TRAC PPO and warmstarted ADAM PPO. No significant benefit is found by warmstarting TRAC PPO compared to not warmstarting it.

## C   Other RL Baselines

While PPO is a widely used policy gradient method in reinforcement learning, it is not the only approach applicable to lifelong RL. Other continual RL methods, such as IMPALA (Espeholt et al., 2018), Online EWC (Schwarz et al., 2018), CLEAR (Rolnick et al., 2019), and Modulating Masks (Nath et al., 2023), are designed to address challenges like catastrophic forgetting in dynamic, nonstationary environments. We incorporated these algorithm implementations adapted from the code from Nath et al. (2023) into our experiments to offer a more comprehensive evaluation. These methods vary in their mechanisms for maintaining task performance over time but may still suffer from plasticity loss in later stages of training.

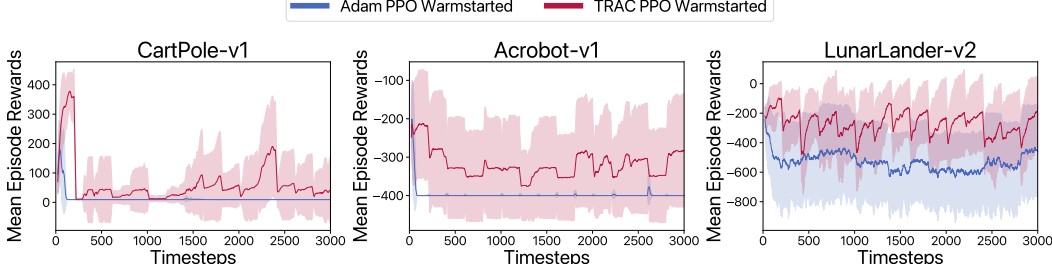

Figure 11: Reward in the lifelong Gym Control environments for CartPole-v1, Acrobot-v1, and LunarLander-v2 with warmstarted TRAC PPO and warmstarted ADAM PPO. TRAC PPO still avoids loss of plasticity and policy collapse.

Table 2: Cumulative sum of mean episode reward over all distributions for ADAM PPO warmstarted and TRAC PPO warmstarted on Procgen, Atari, and Gym Control environments. Rewards are scaled by $10^5$; higher is better.

| Environment | ADAM PPO | TRAC PPO (Ours) |
|---|---|---|
| Starpilot | 3.0 | **10.2** |
| Dodgeball | 1.2 | **2.5** |
| Chaser | 1.3 | **1.6** |
| Fruitbot | −0.4 | **0.6** |
| CartPole | 4.6 | **22.8** |
| Acrobot | −142.9 | **−114.5** |
| LunarLander | −190.7 | **−97.3** |
| Atari6 | 16.7 | **72.2** |
| Atari9 | 34.6 | **80.6** |

**Mitigating plasticity loss across policy methods:** Figure 12 demonstrates the performance of various continual RL methods when paired with ADAM and TRAC optimizers. The results indicate that when using ADAM, methods like IMPALA, Online EWC, CLEAR, and Modulating Masks exhibit a noticeable decline in performance over time due to plasticity loss, particularly in later levels of the Procgen environments. In contrast, pairing these methods with TRAC instead of ADAM leads to significant improvements, mitigating plasticity loss and enhancing reward performance across subsequent distribution shifts.

To quantify these improvements, Figures 13, 12 present the average normalized rewards over five seeds and 120M timesteps for each method across four different Procgen environments: Starpilot, Dodgeball, Chaser, and Fruitbot. Across all environments, methods that use TRAC outperform their Adam-based counterparts, consistently maintaining higher rewards over time.

On average, across the Procgen environments, TRAC led to performance improvements over ADAM by the following margins: **21.83%** for IMPALA, **15.86%** for Online EWC, **14.41%** for CLEAR, and **10.14%** for Modulating Masks.

**General Applicability of TRAC:** It is important to highlight that TRAC is orthogonal to the learning or policy algorithms themselves. It can be seamlessly integrated into various reinforcement learning architectures by simply replacing their optimizer (e.g., ADAM or RMSPROP). Our results demonstrate that TRAC enhances performance across different algorithms and environments, consistently outperforming ADAM in mitigating plasticity loss.

## D   Gravity Based Distribution Shifts

One method to introduce distribution changes in reinforcement learning environments is by altering the dynamics Mendez et al. (2020), such as adjusting the gravity in the CartPole environment. In this set of experiments, we manipulate the gravity by a magnitude of ten, randomly adding noise for one distribution shift, and then inversely, dividing by ten and adding random noise for the next shift. This process continues throughout the experiment.

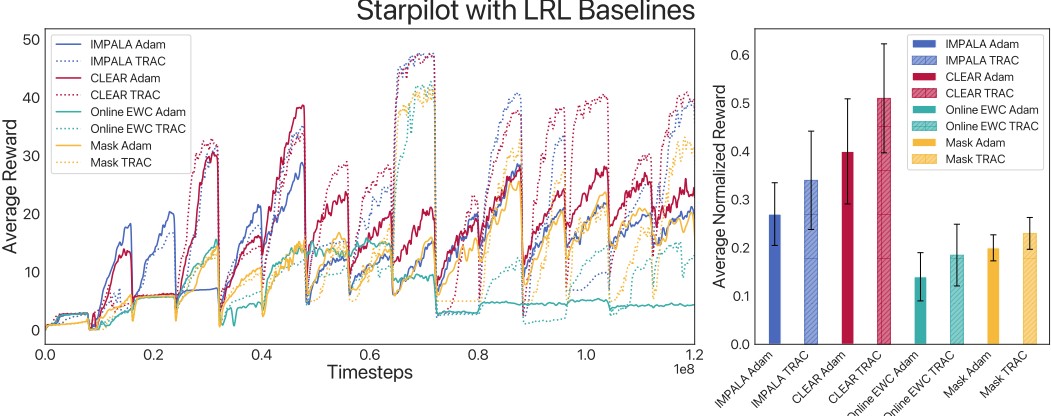

Figure 12: Performance comparison between Adam-based and TRAC-based continual RL methods (IMPALA, Online EWC, CLEAR, Modulating Masks) in Starpilot. While ADAM suffers from plasticity loss in later levels, TRAC effectively mitigates this and maintains better performance over distribution shifts. For clarity, standard deviation fills are omitted here but included in the bar plot.

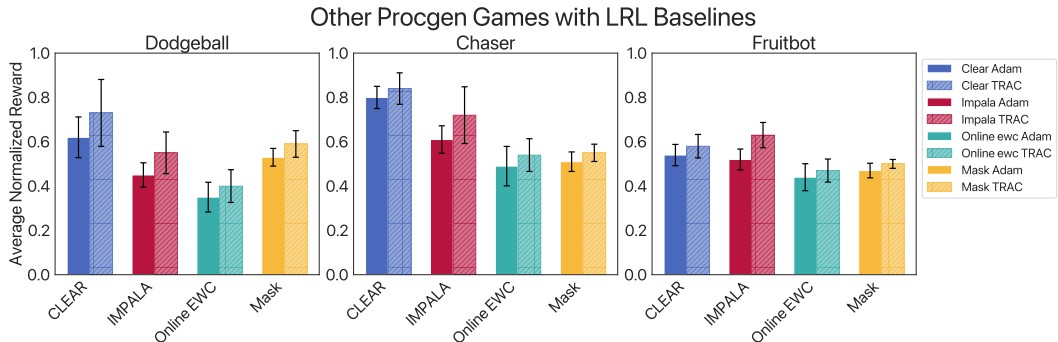

Figure 13: Average normalized rewards over five seeds and 120M timesteps for Dodeball, Chaser, and Fruitbot. Each method (IMPALA, Online EWC, CLEAR, and Modulating Masks) is evaluated using both Adam and TRAC. TRAC consistently outperforms ADAM across all methods and environments, with improvements ranging from 10% to 21%.

Our observations suggest that ADAM PPO is robust to such dynamics-based distribution shifts, as shown in Figure 14. This indicates that while ADAM PPO implicitly models the dynamics of the environment well—where changes in dynamics minimally impact performance—it struggles more with adapting to out-of-distribution observations such as seen in the main experiments (Figure 6) and in the warmstarting experiments (Figure 11).

## E LayerNorm, Plasticity Injection, and Weight Decay

To evaluate TRAC alongside other methods that aim to mitigate plasticity loss, we compare it against LayerNorm (Lyle et al., 2023), Plasticity Injection (Nikishin et al., 2024), and tuning weight decay (Lyle et al., 2024).

As discussed in Section 5, we confirm that both layer normalization and plasticity injection (applied at the start of every distribution shift) (Nikishin et al., 2024; Lyle et al., 2023) are effective in reducing plasticity loss (Figure 15). While these methods help slow the decline in performance due to plasticity loss, TRAC consistently outperforms them across the three Gym Control environments. Importantly, because TRAC is an optimizer, it can be combined with layer normalization, and doing so resulted in the best performance gains in our Control setups.

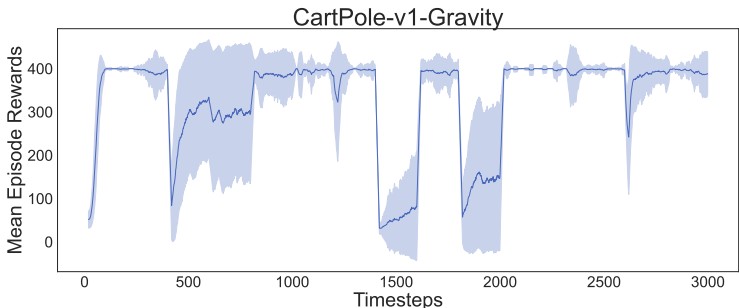

Figure 14: Mean Episode Reward for ADAM PPO on CartPole-v1 with varying gravity. ADAM PPO demonstrates robust policy recovery across most gravity-based distribution shifts.

**Tuning weight decay:** In addition to LayerNorm and Plasticity Injection, we also evaluated the effects of tuning weight decay using PyTorch's AdamW optimizer. We conducted a hyperparameter sweep across three control environments with 15 seeds for each of the following weight decay values: 0.0001, 0.001, 0.01, 0.1, 1.0, 5.0, 10.0, 15.0, and 50.0. Figure 16 presents the average normalized reward for each weight decay value over 15 seeds and 3000 timesteps, compared to TRAC.

The results indicate that while tuning weight decay with Adam does provide some benefit, these values consistently underperform in comparison to TRAC across all three control environments. Figure 16 plots the performance of the best-performing weight decay value with Adam over 10 distribution shifts in the control environments. We observe that weight decay values are highly sensitive to the specific environment and the nature of the distribution shift.

Interestingly, in our initial experiments, we set the weight decay to zero, yet TRAC still outperformed Adam with various weight decay values. This suggests that while weight decay can mitigate plasticity loss to some extent, it does not match the overall effectiveness of TRAC.

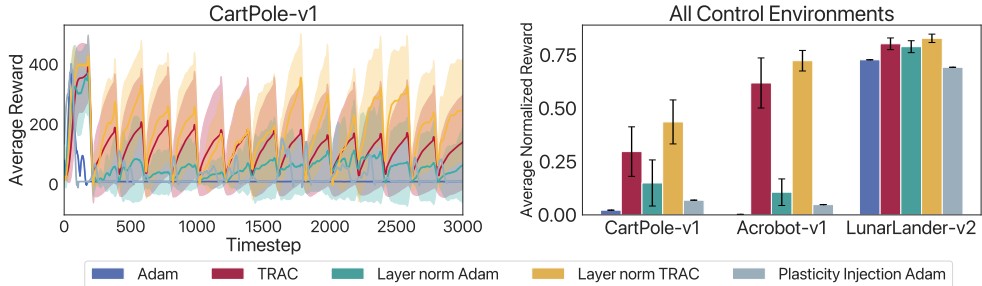

Figure 15: Performance comparison of plasticity loss mitigation techniques across Gym Control environments. Both layer normalization and plasticity injection reduce plasticity loss when applied with ADAM. TRAC outperforms both layer norm ADAM and plasticity injection ADAM, with the combination of layer norm and TRAC achieving the highest performance.

## F  Scaling-Value Convergence

As discussed in the algorithm section (see Section 3), TRAC operates as a meta-algorithm on top of a standard optimizer, denoted as BASE. The crucial component of TRAC involves the dynamic adjustment of the scaling parameter $S_{t+1}$, managed by the tuner algorithm (Algorithm 2). This parameter is data-dependent and typically ranges between $[0, 1]$. The weight update $\theta_{t+1}$ is consequently defined as a convex combination of the current optimizer's weight $\theta_t^{\text{BASE}}$ and a predetermined reference point $\theta_{\text{ref}}$.

This section presents the convergence behavior of the scaling parameter $S_{t+1}$ across different environments, analyzed through the mean values over multiple seeds.

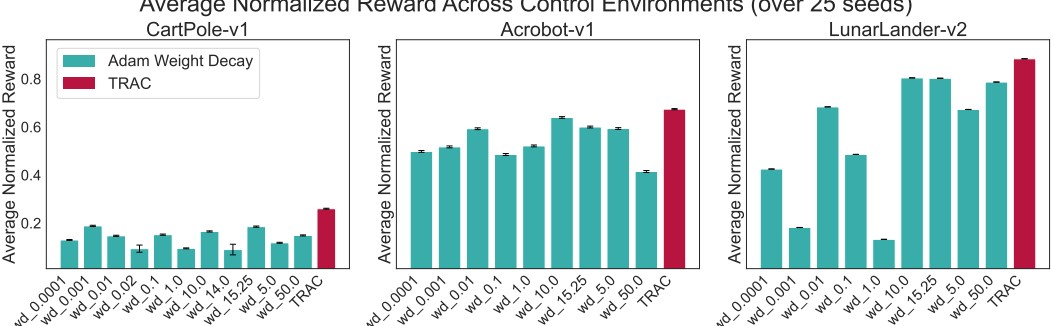

Figure 16: Effect of weight decay on performance in the three Gym Control environments. Bar plots show the average normalized rewards over 25 seeds for different weight decay values using ADAM across 3000 timesteps, compared to TRAC with no weight decay.

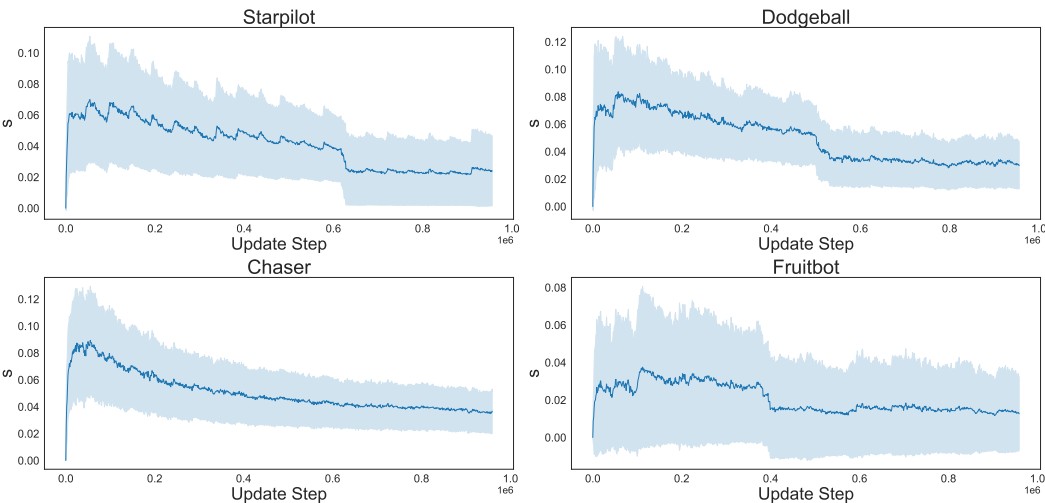

Figure 17: Convergence of the scaling parameter $S_{t+1}$ in the Procgen environments.

The convergence of the scaling parameter $S_{t+1}$ observed across the Procgen and Gym Control environments, as depicted in Figures 17 and 19, reflects a good scaling value that effectively determines the strength of regularization towards the initialization points, yielding robust empirical outcomes in lifelong RL settings. Interestingly, in Procgen environments, this converged scaling value exhibits consistency across various games, typically hovering between 0.02 and 0.03, as shown in Figure 17. In contrast, in the Gym Control environments, the scaling values are lower, ranging between 0.005 and 0.01, as illustrated in Figure 19.

## G   Comparison to MECHANIC

In our analysis, we extend the examination to other OCO-based optimizers within the lifelong RL setup. Table 3 presents a comparative assessment of TRAC PPO and MECHANIC PPO (Cutkosky et al., 2023) for the lifelong Gym Control tasks (with 300 seed runs). The p-values were calculated using two-sample t-tests to test the hypothesis that the means between TRAC and MECHANIC are the same (Null Hypothesis, $H_0$) against the alternative hypothesis that they are different (Alternative Hypothesis, $H_1$). The results indicate that while MECHANIC effectively mitigates plasticity loss and adapts quickly to new distribution shifts, it slightly underperforms in comparison to TRAC.

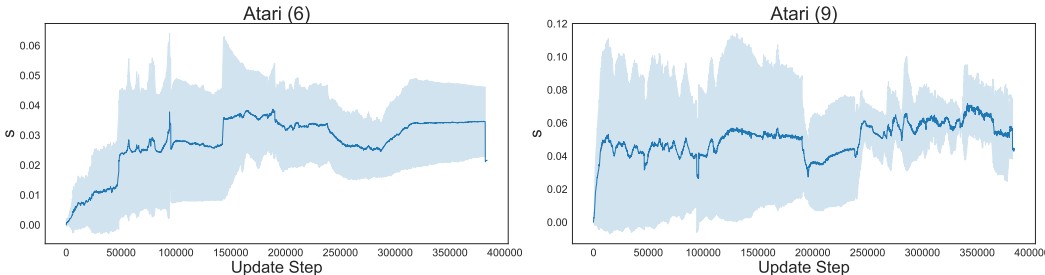

Figure 18: Evolution of the scaling parameter $S_{t+1}$ in the Atari environments. Here we don't see a meaningful convergence of $S_{t+1}$.

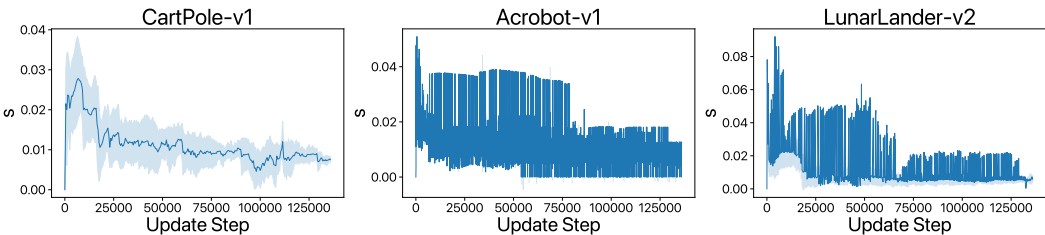

Figure 19: Convergence of the scaling parameter $S_{t+1}$ in the Gym Control environments.

# H   Experimental Setup

**Procgen and Atari Vision backbone**   For both the Atari and Procgen experiments, the Impala architecture was used as the vision backbone. The Impala model had 3 Impala blocks, each containing a convolutional layer followed by 2 residual blocks. The output of this is flattened and connected to a fully connected layer. The impala model parameters are initialized using Xavier uniform initialization.

**Policy and Value Networks**   Across all experiments—including Gym Control, Atari, and Procgen—the policy and value functions are implemented using a multi-layer perceptron (MLP) architecture. This architecture processes the input features into action probabilities and state value estimates. The MLP comprises several fully connected layers activated by ReLU. The output from the final layer uses a softmax activation.

**TRAC**   TRAC, for all experiments, was implemented using the same experiment-specific baseline architectures and baseline optimizer. For the Procgen and Atari experiments, the base ADAM optimizer was configured as the same as baseline, with a learning rate of 0.001, and for the Gym Control experiments, a learning rate of 0.01 was used. Both learning rates were tested for all experiments and found to have negligible differences in performance outcomes. Other than the learning rate, we use the default ADAM parameters, including weight decay and betas, followed by the specifications outlined in the PyTorch Documentation.[5]

The setup for TRAC included $\beta$ values for adaptive gradient adjustments: 0.9, 0.99, 0.999, 0.9999, 0.99999, and 0.999999. Both $S_t$ and $\varepsilon$ were initially set to $(1 \times 10^{-8})$. Modifications were made to a PyTorch error function library, which accepts complex inputs to accommodate the necessary computations for the imaginary error function. This library can be found at Torch Erf GitHub.[6]

**Distribution Shifts**   In the Atari experiments, game environments were switched every 4 million steps. The sequence for games with an action space of 6 included "BasicMath", "Qbert", "SpaceInvaders", "UpNDown", "Galaxian", "Bowling", "Demonattack", "NameThisGame", while games with an action space of 9 included "LostLuggage", "VideoPinball", "BeamRider", "Asterix", "Enduro", "CrazyClimber", "MsPacman", "Koolaid".

---

[5]https://pytorch.org/docs/stable/generated/torch.optim.Adam.html
[6]https://github.com/redsnic/torch_erf

Table 3: Performance comparison between TRAC and MECHANIC across three Gym Control environments. The mean, standard error, and p-values reflect the performance over multiple runs, with bolded values highlighting TRAC's superior results.

| Task | Method | Mean | Std Error | p-value |
|---|---|---|---|---|
| LunarLander-v2 | TRAC | **0.6018** | 0.0036 | 0.0000 |
| | Mechanic | 0.5755 | 0.0027 | |
| CartPole-v1 | TRAC | **0.3518** | 0.0244 | 0.0021 |
| | Mechanic | 0.3008 | 0.0230 | |
| Acrobot-v1 | TRAC | **0.7044** | 0.0221 | 0.0000 |
| | Mechanic | 0.6396 | 0.0239 | |

Table 4: PPO Parameters for Atari, Procgen, and Gym Control Experiments

| Parameter | Atari | Procgen | Control |
|---|---|---|---|
| Steps per update | 2,000 | 1,000 | 800 (2 episodes with 400 steps) |
| Batch size | 250 | 125 | 32 |
| Epochs per update | 3 | 3 | 5 |
| Epsilon clip for PPO | 0.2 | 0.2 | 0.2 |
| Value coefficient | 0.5 | 0.5 | 0.5 |
| Entropy coefficient | 0.01 | 0.01 | 0.01 |
| Base Optimizer | ADAM (LR: 0.001) | ADAM (LR: 0.001) | ADAM (LR: 0.01) |
| Architecture | Impala + MLP | Impala + MLP | MLP |

For Procgen experiments, individual game levels were sampled using a seed value as the *start_level* parameter, which was incremented sequentially to generate new levels. Each new environment was introduced every 2 million steps.

In the Gym Control experiments, each observation dimension was randomly perturbed by a value ranging from 0 to 2. This perturbation was constant for 200 timesteps, after which a new perturbation was applied, effectively switching the environmental conditions every 200 steps.

**Statistical Significance**    The Procgen and Atari experiments were conducted with 8 seeds/runs, while the Gym Control experiments utilized 25 seeds/runs (with the exception of the Mechanic experiments in Table 3 which utilized 300 seeds). The exception was in the $L_2$ initialization experiments, which used 15 seeds/runs per regularization strength. In Figures 4, 5, 6, 7, 9, 10, 11, 15, 12, 14, the plotted lines represent the mean of all of the mean episode rewards from the different seeds/runs, and the shaded error bands indicate the standard deviation of all of the mean episode rewards from the different seeds/runs.

**Compute Resources**    For the Procgen and Atari experiments, each was allocated a single A100 GPU, typically running for 3-4 days to complete. The Gym Control experiments were conducted using dual-core CPUs, generally concluding within a few hours. In both scenarios, an allocation of 8GB of RAM was sufficient to meet the computational demands.

