# OpenReview forum: "Fast TRAC: A Parameter-Free Optimizer for Lifelong Reinforcement Learning"
_NeurIPS.cc/2024/Conference — NeurIPS 2024 poster_

### Official Review · Reviewer_Jcag · 2024-06-26

**Soundness:** 2
**Presentation:** 2
**Contribution:** 2
**Rating:** 5
**Confidence:** 2

**Summary:**

This paper proposes a new method for avoiding plasticity loss in 'lifelong RL', a setting in which distribution shift occurs during RL training. Here, the setting is split into a number of discrete tasks which change at a fixed interval. The method involves including a penalty on updates, rooted in convex optimization, which encourages movement back to a reference parameterisation. Upon testing, they find that performance sees drastic improvement over two baselines, and does not suffer from the effects of plasticity loss in the same way.

**Strengths:**

- Results are clearly drastically improved with the introduction of PACE, compared to the algorithms which it is benchmarked against *in the main paper*.
- The work seems well motivated, with the section on Lifelong RL outlining the justification for why this is a useful direction to study.
- The work was grounded in theory. The justification for design decisions in the context of OCO literature clearly enhanced the method.
- The paper was clearly written in many places, with clarity lost in only a small number of places (see weaknesses).
- I have personally not seen previous work which pursues this direction of hyperparameter-free plasticity loss mitigation, particularly in this way. I think this is project has good novelty.
- The results are demonstrated on a wide array of environments, which I believe to be a sensible distribution for evaluation in this setting.
- The figures were great for offering intuition about what was happening in the method, and the problem setting.
- The extra analysis in the appendix was quite enlightening and was appreciated.

**Weaknesses:**

There are a few key weaknesses that I think really hold this paper back, which I will detail below. However, I am willing to increase my score based on rebuttals to the following weaknesses and questions.
- The key thing for me is that it seems Mechanic, in the appendix, performs *very* similarly to PACE (and well within the standard error, making it hard to make any claims about statistical significance - undermining the claim that Mechanic 'slightly underperforms in comparison to PACE'). While I am unfamiliar with this preexisting method, I think it would be good to have included this result into the main paper, with tests across more environments. I appreciate that the discussion mentions that Mechanic has worse guarantees, but empirical demonstration for why PACE is better is pretty necessary given that Mechanic has been included.
- I also think there are missing comparisons to other methods referenced in the paper, which are designed to mitigate plasticity loss. While I appreciate these may not come under the 'hyperparameter-free' setting, comparison to other methods like plasticity injection would be beneficial for showing why PACE is worth using instead.
- In places, the method section was quite confusing. Making the algorithms and subsequent text more concise would be useful as it took me quite a few reads to properly parse and understand.
- The way the results are compared in the table is quite disingenuous, since I believe (particularly in Atari) the results are warped by performance in a very small number of the selected games which had a higher reward scale. This is made clear by the return curves, which show the same story as the table but in an arguably more demonstrative way. It could be worth either normalising scores between each environment, or doing away with the tables entirely.
- As a small nitpick, it felt like lifelong RL is introduced far before its accompanying reference, making it sound like a new setting created for the paper rather than a term used in this context commonly.
- The opening paragraph read in quite a different tone than the rest of the paper. It comes across in a slightly unserious manner that I think detracts from the rest of the paper. Having real-world motivation of the problem setting is very good, but this could have been written better (I appreciate this could be quite a personal opinion, and so did not incorporate it into my score).

**Questions:**

- As mentioned earlier, could you please offer a more rigorous comparison with Mechanic (or explain in further detail why PACE is much better, since the difference seems so far very marginal)?
- How much weight decay was used for Adam? Since this method is very closely aligned to weight decay, and the S value was usually *roughly* a constant scale, could tuning the extent of weight decay provide the same benefit as PACE? As pointed out by [1], weight decay and L2 regularisation are marginally different when using Adam. I appreciate this inroduces an additional tunable hyperparemeter, which is against the spirit of the paper.



[1] Kumar, Saurabh et al. “Maintaining Plasticity in Continual Learning via Regenerative Regularization.” (2023).

**Limitations:**

The authors discuss some limitations of their work, such as limited analysis with respect to different algorithms and a brief discussion of Mechanic, another method which performs very similarly to PACE. While I have briefly discussed additional results in regards to limitations, I feel the authors are upfront about a lot of the limitations of their work and that these are not overly limiting to the work.

---

> ### Author Rebuttal · Authors · 2024-08-07
>
> We thank you so much for your insightful review, feedback, and opportunity to improve our work!
>
> ## PACE vs. Mechanic
>
> Firstly, we address your feedback by conducting a more rigorous evaluation of Mechanic across our benchmarks. Implementing your suggestion has substantiated our claim that PACE performs slightly better empirically.
>
> ### Results
> We conducted additional experiments, running 10 seeds for Mechanic on Procgen and Atari environments and 25 seeds on Control environments. We show the average normalized reward over the entire lifelong run for each environment for PPO when using both PACE and Mechanic in Supplementary PDF Figure 2.
>
> Following your suggestion, all results below are average normalized rewards (normalized scores between each environment, and for atari, normalized for each game; see more details on this in the Author Rebuttal).
>
>
> **Procgen Results for Mechanic vs PACE over 10 seeds**
> | Mechanic Starpilot | PACE Starpilot | Mechanic Dodgeball | PACE Dodgeball | Mechanic Chaser | PACE Chaser | Mechanic Fruitbot | PACE Fruibot
> | -------- | -------- | -------- | -------- | -------- | -------- | -------- | -------- |
> | 0.49±0.0494     | **0.53±0.0468**     | 0.50±0.1267     |**0.51±0.1258**     |0.41±0.1232     |**0.52±0.0759**     |0.57±0.1356     |**0.65±0.1258**     |
>
> **Control Results for Mechanic vs PACE over 25 seeds**
>
> | Mechanic CartPole | PACE CartPole | Mechanic Acrobot | PACE Acrobot | Mechanic LunarLander | PACE LunarLander
> | -------- | -------- | -------- | -------- | -------- | -------- |
> | 0.31±0.1169     | **0.32±0.1092**     | 0.69±0.1150     |**0.72±0.1153**     |0.52±0.0292     |**0.54±0.0165** |
>
> **Atari Results for Mechanic vs PACE over 10 seeds**
>
> | Mechanic Atari (6) | PACE Atari (6) | Mechanic Atari (9) | PACE Atari (9)
> | -------- | -------- | -------- | -------- |
> | 0.45±0.04451     | **0.48±0.0389**     | 0.55±0.05630     |**0.58±0.0407**|
>
>
>
> ### Conclusion
> **While Mechanic effectively combats plasticity loss and performs similarly to PACE, it consistently underperforms PACE by a slight margin across all settings.** *This result may align with the theoretical expectation that our scaling tuner provides better OCO performance guarantees.*
>
> ### Insight
>
> While Mechanic is also a parameter-free optimizer, it has only been evaluated in the context of stationary supervised DL. By demonstrating that Mechanic performs well in a variety of non-stationary RL settings, we strengthen our claim that parameter-free optimization deserves more attention in the context of lifelong RL. We believe these results further validate the insights derived from PACE:
> - parameter-free OCO is all about better regularization and
> - better regularization mitigates loss of plasticity.
>
> We are happy to include these additional Mechanic results in our main experiments.
>
>
> ## Weight Decay Tuning
>
> We appreciate your question regarding the tuning of weight decay with Adam, as explored by Lyle et al. 2023 for mitigating plasticity loss.
>
> * In our experiments, weight decay was set to 0.
> *  Following your suggestion, we conducted a HP sweep using PyTorch’s AdamW optimizer across three control environments with 15 seeds for each of the following weight decay values: `0.0001, 0.001, 0.01, 0.1, 1.0, 10.0, 15.0, 5.0, and 50.0`.
> *  Our findings show varying weight decay values do offer benefits (Supplementary PDF Figures 4a-c), but do not outperform PACE with no weight decay.
> *  Like L2 regularization strengths, the best weight decay values are environment-specific and sensitive to distribution shifts (see Supplementary PDF Figure 4d), complicating the presetting of these values without a prior hyperparameter sweep or knowledge of expected distribution shifts.
>
>
> ## Plasticity Loss Mitigation Baselines
> In appreciation of your suggestion, we have included plasticity injection (following the simple methodology laid out in Nikishin et al.) as a baseline in our control environments.
>
> * We inject plasticity at the start of every distribution shift.
> * While plasticity injection does help mitigate plasticity loss, it is outperformed by PACE across the 3 control settings (Supplementary PDF Figure 3b).
>
> Additionally, we included another plasticity loss mitigation technique, layer normalization, as suggested by Reviewer eCfr.
> * While layer normalization improves the performance of baseline Adam, it is outperformed by PACE across the 3 control settings.
> * However, we also find that layer normalization with PACE improves over PACE without layer normalization (Supplementary PDF Figure 3b).
>
>
> ## Minor: Reward Normalizizaton
> Thank you for pointing out that our results can be made more clear by normalizing the scores in Table 1 (we will follow the normalization procedure in the Author Rebuttal).
>
> Please let us know if you have any other concerns/comments, and we'll be glad to address those!

---

> > ### Comment · Reviewer_Jcag · 2024-08-07
> >
> > Dear authors,
> >
> > Thank you for your rebuttal.
> >
> > # Extra Mechanic Experiments
> >
> > These provided results reaffirm my point. It is important to highlight that, generally, bolding is reserved for **statistically significant results**. In your provided results, every single value for Mechanic is within the standard (error? deviation? sorry I'm not sure what your intervals are) of PACE, suggesting the results are not statististically significant. In some cases, the differences are on the order of 0.01 with error bars in the range 0.1 - ten times the difference!
> >
> > # Weight Decay
> >
> > Thank you for running these experiments. It does seem like PACE has improvements over Adam + WD, but it is interesting how the weight decay has improved the performance of Adam.
> >
> > # Plasticity Loss Mitigation
> >
> > Thank you for running these additional baselines. In Cartpole and LunarLander I have concerns about statistical significance between Adam + LN and vanilla PACE, but it is interesting how PACE + LN seems genuinely better than PACE.
> >
> > Overall, while your weight decay and plasticity loss experiments are certainly interesting, my primary concern still remains on the performance differential between PACE and Mechanic. In fact, I would say these results have made me less sure that PACE offers any benefits over Mechanic. Are there any other considerations that can differentiate between the two, such as difference in compute/speed?
> >
> > As it stands, I am not willing to change my score and still recommend borderline rejection of this paper as my principal concern remains unresolved.

---

> > > ### Author Response · Authors · 2024-08-08
> > >
> > > Thank you for your quick response to our experiments!
> > >
> > > We agree with you and acknowledge that more seeds are needed for the precise comparison of PACE and Mechanic. However, we would like to emphasize that **the main message of this paper is the effectiveness of parameter-free OCO for mitigating plasticity loss.** In this regard, PACE vs Mechanic is somewhat a secondary consideration. It's important to note that
> > > 1. Mechanic is designed for the classical DL training regime, and hasn't been applied to any lifelong learning setting before.
> > > 2. Both PACE and Mechanic are better than existing approaches in lifelong RL (like $L_2$ regularization and plasticity injection) which require a hyperparameter tuning oracle that's impossible in practice.
> > >
> > > As for how exactly PACE compares to Mechanic, our current experiments show that PACE is better, although we agree with you that more seeds are needed to strengthen this argument (we'll do an even more rigoruous comparison in the camera-ready revision). This is also consistent with the OCO theory: Zhang et al (2024b) showed that the underlying 1D tuner of PACE has a better OCO regret bound (actually, optimal) than the tuner of Mechanic, although the improvement is on the log factor and multiplying constant (instead of the $\tilde O(\cdot)$). In other words, PACE is theoretically better by a slight margin, and it should be *very* hard to improve even further.
> > >
> > > Regarding our contribution, we would also add that our paper is not an "A+B" type of work, where one takes a method A and blindly applies it to a new domain B, hoping it "beats" the benchmark there. Instead, our paper reveals some (in our opinion) neat and surprising insights.
> > > - For the lifelong RL community, we show that: (1) Parameter-free OCO provides better regularization than other common approaches (e.g., weight decay). (2) Better regularization mitigates loss of plasticity.
> > > - For the parameter-free OCO community, we show that one could jump out of the typical application regime (DL training), and use the intrinsic connection to regularization to achieve interesting results. In particular, DL training often involves iid data, where applying OCO is somewhat an overkill. Lifelong RL concerns complicated nonstationary data generation mechanisms, which means that the full capability of OCO algorithms can be better exploited.
> > >
> > >
> > > In summary, our intended primary contribution lies in demonstrating that parameter-free OCO mitigates plasticity loss in lifelong RL. With the camera-ready version, we will emphasize how PACE, alongside Mechanic, substantiates this point, clarifying the benefit of parameter-free OCO in lifelong RL.
> > >
> > >
> > > Thanks again! And please let us know if there's anything else we can do to change your mind.

---

> ### Author Response · Authors · 2024-08-10
> **More PACE and Mechanic seeds for Gym Control experiments**
>
> Dear Reviewer,
>
> Following up on our previous comment regarding the statistical significance of our results, we have conducted additional experiments on the Gym Control suite, which are more feasible, quicker, and efficient to run on CPUs than the other experiments. Specifically, we increased the sample size to 300 seeds for CartPole, Acrobot, and LunarLander. While our rebuttal initially reported the mean and standard deviations of the seed runs, we now provide a more stringent analysis, including the mean, standard error, and p-values, to make our claim as strong as possible. P-values were calculated using two-sample t-tests. For each control task. we tested the hypothesis that the means between Mechanic and PACE are the same ($\mu_{\text{PACE}} = \mu_{\text{Mechanic}}$)) (Null Hypothesis, H0) against the alternative that they are different (Alternative Hypothesis, H1). P-values here are helpful to quantify the probability of observing the data under the assumption that the null hypothesis is true.
>
>
> ### Results:
>
> | Task             | Method    |  Mean   | Std Error | p-value |
> |------------------|-----------|--------|-----------|---------|
> | **LunarLander-v2**| PACE       | 0.6018 | 0.0036    | 0.0000  |
> |                  | Mechanic   | 0.5755 | 0.0027    |         |
> | **CartPole-v1**  | PACE       | 0.3518 | 0.0244    | 0.0021  |
> |                  | Mechanic   | 0.3008 | 0.0230    |         |
> | **Acrobot-v1**   | PACE       | 0.7044 | 0.0221    | 0.0000  |
> |                  | Mechanic   | 0.6396 | 0.0239    |         |
>
> **These results support our argument that PACE outperforms Mechanic in the Gym Control tasks.** The p-values here,  significantly below the commonly accepted threshold of 0.05, strongly reject the null hypothesis in favor of the alternative. This rejection supports our claim about the efficacy of PACE over Mechanic. Additionally, the relatively low standard errors associated with the mean scores for each method further indicate a high level of precision in our results, reinforcing the robustness of our findings.
>
> Even with these results, we reiterate that our primary contribution remains the demonstration that parameter-free OCO helps mitigate the loss of plasticity in lifelong RL.
>
> **Please let us know if this has changed your mind or if you have any other concerns/questions that were not addressed here.**

---

> > ### Comment · Reviewer_Jcag · 2024-08-12
> >
> > Dear authors,
> >
> > Trying to present the main message of your paper as anything besides the algorithm seems to me to be a mischaracterisation, on the basis that your introduction features a contribution section stating that the paper has two contributions:
> > - Algorithm (i.e. PACE)
> > - Experimental (i.e. showing that PACE outperforms standard PPO)
> > To suggest that the paper is not an algorithmic contribution, but one of analysis, seems questionable and would require significant restructuring of the paper. GIven the contribution set out in your introduction is an algorithmic one, it seems important to ensure that your algorithm outperforms the baseline with significance.
> >
> > Though your additional experiments required a large number of seeds, I agree that they do seem to demonstrate some significance for the claim that PACE > Mechanic. As such, I have decided to increase my score to reflect this, increasing to borderline acceptance. However, I maintain my low confidence since this has only been shown in a small number of the environments and the improvement seems to be **very** marginal, and as such I still find it very hard to make strong claims about the performance or usefulness of PACE. It also feels that a potentially large amount of restructuring will be required to the paper to cover the necessary changes discussed here, which may require extra reviewing.
> >
> > Thank you for your engagement in discussions and running additional experiments; I appreciate that this can be very time consuming during a busy time, but I have felt this conversation has been enlightening regarding your paper.

---

> > > ### Author Response · Authors · 2024-08-12
> > > **Thank you**
> > >
> > > Dear Reviewer,
> > >
> > > Thank you for recognizing our additional experiments showing PACE is better than Mechanic and increasing your original score! We appreciate your constructive comments and will polish the camera-ready version to make the presentation of our contribution fair and easy to understand. We are grateful for your engagement in our discussions.

---

### Official Review · Reviewer_Jqmp · 2024-07-12

**Soundness:** 3
**Presentation:** 1
**Contribution:** 3
**Rating:** 6
**Confidence:** 3

**Summary:**

Many methods have been proposed recently for continual learning that can maintain their plasticity (e.g., through resetting or regularization towards initialization). There is, however, one critical limitation of those methods, which is their hyper-parameter tuning. Although those methods are designed for lifelong learning that can potentially last forever, their performance is highly dependent on the hyperparameter choices. This paper attempts to address this gap in research by introducing a tuning-free approach inspired by online convex optimization for continual learning by incorporating multiple ideas from online convex optimization research and showing that their method PACE is insensitive to hyperparameter choices and can work in a wide range of problems.

**Strengths:**

This paper addresses an important limitation of existing continual learning methods, which is their dependence on hyperparameter tuning. The paper presents a novel approach that requires no tuning for hyperparameters and demonstrates the effectiveness of their method on a range of problems. The implications of tuning-free continual learning are immense and can advance future research on continual learning.

**Weaknesses:**

- Writing needs to be improved. Several parts of the paper seem incomplete, which may require significant restructuring of the paper.
  - The information on online convex optimization doesn’t fully introduce relevant concepts used in the algorithm. Thus, they either need to be removed, and the paper would rely on general intuition to motivate their method and explain how it’s generally related to OCO or to introduce the method in terms of OCO rigorously. The paper, as it stands, does not do either option.
  - The algorithm is not fully explained. For example, how the three components work together is not explained, and the paper relies entirely on citing the papers in which they incorporate their ideas in the algorithm. This prevents the paper from being self-contained. Additionally, what does the dot product between the instantaneous gradient $g_t$ and $\theta_t- \theta_{\text{ref}}$ represent? This was never explained, although it’s a crucial part of the algorithm. The following questions should be answered by reading the algorithm and its explanation: 1) the role of the erfi function and 2) why are we keeping track of the first and second moments of the dot product between the current gradient and $\theta-\theta_{\text{ref}}$? 3) how does the additive aggregation work?
  - Writing should be more focused on what is crucial for following ideas explained in the paper. For example, the paper explained static regret and then mentioned that discounted regret is more suitable only in one sentence without a proper introduction. Similarly, the paper explains the minimax algorithms for static regret but never explains how parameter-free methods are different. I don’t thinking static regret or minmax formulation is helping in the flow of the paper, instead it may hurt it since it distracts the reader from the main message without mentioning the importance of parts for following along.
- The effectiveness of the three components of the proposed method has not been investigated. Specifically, an ablation study is needed to understand the role of each component and how they interact with each other. For example, the authors might add an ablation study where one component is removed at a time or one component is added at a time.

**Minor issues and suggestions:**
- The following claim in the conclusion section is unsubstantiated: “PACE’s success leads to a compelling takeaway: empirical lifelong RL scenarios may exhibit more convex properties than previously appreciated and might inherently benefit from parameter-free OCO approaches”. The effectiveness of your method doesn’t mean it works because the problem has convex properties. More work is needed to support such a claim and find direct evidence.
- I suggest the method may be called tuning-free, not parameter-free, to enhance readability for a wide range of audiences.
- “From the optimization perspective, the above issues might be attributed to the lack of stability under gradient descent.” -> This requires a citation. Also, what does stability mean in this context?
- line 78: $T$ is not defined. Also, it's doubly used to mean a different thing in line 120.
- Notation in equation 1 is not fully defined (e.g., what is $o(T)$?)
- The word "conservatism” is used loosely without properly defining what it means.
- I believe the “Lifelong vs. Continual” paragraph needs to be dropped. The two words mean the same thing. There is no need to use them in different contexts to mean different things. This would lead to unnecessary division in using the word. Further, I don't think Abbas et al. (2023) focused on backward transfer; instead, they focused on the loss of plasticity.
- the axis in Figure 4 should be the “average episodic return” or "average cumulative rewards" not “average episode rewards”.

**Questions:**

- What is a round? (line 121)
- No definition of discounted regret has been given. How is it defined?
- “Algorithms there mostly follow  the same principle as in the stationary setting, just wrapped by loss rescaling (Zhang et al., 2024a)." What does this mean?
- What does the distribution shift (from 1 to 10) mean in Figure 7 on the x-axis of the first plot?
- Why is the cumulative sum of rewards used while the authors mentioned that discounted regret is suitable for lifelong RL?
- The paragraph starting at line 126 is unclear. What does uncertainty set $\mathcal{U}$ mean or represent?

**Limitations:**

The authors adequately addressed the limitations of their work. However, I think since PACE exhibits suboptimal performance in the early stages of deployment, it might suggest that it doesn’t also adapt to a fast-changing sequence of tasks.

---

> ### Author Rebuttal · Authors · 2024-08-07
>
> Thank you for your very thorough review! Especially, we appreciate your valuable comments on the writing and organization of this paper. We'll incorporate those in the camera-ready revision.
>
> Regarding your questions: we realized that we went a bit too fast when introducing the algorithm. So first of all we would like to clarify its main idea, as well as the role of each component.
>
> ## Idea of PACE
>
> The overall structure of PACE results from a decade of research on parameter-free OCO. The latter as a branch of optimization theory concerns proving regret bounds under convex losses, which has a similar spirit as our problem but very much idealized (e.g., there isn't any notion of plasticity etc).
> - The central component of PACE is the direction-magnitude decomposition from (Cutkosky & Orabona, 2018), whose main idea is to apply a carefully designed 1D algorithm (parameter-free tuner) on top of a base optimizer to adjust how far the iterates drift away from the initialization. With the right tuner (Alg 2), this essentially acts as a data-dependent regularizer. It helps mitigating plasticity loss due to the high plasticity associated with the initialization (around the origin; see the insights of Abbas et al, 2023).
> - The tuner we use is based on the erfi function, due to (Zhang et al, 2024b). The design of this tuner is a long story (solving a PDE called the Backward Heat Equation), but intuitively, the idea is that the shape of the erfi function trades off the distance of the iterates from the origin and from the empirical optimum. Such a tradeoff is mathematically optimal in a quite strong sense.
> The tuner takes the inner product $\langle g_t,\theta_t-\theta_{\mathrm{ref}} \rangle$ as input, which is essentially the projection of the gradient along the update direction (since tuner controls the magnitude rather than direction, it only cares about the gradient component on the direction of $\theta_t-\theta_{\mathrm{ref}}$).
> - As in much of practical model training, the tuner requires discounting. To automatically select the discount factor, we use the additive aggregation technique from (Cutkosky, 2019). The argument is very simple: if we *sum up* two parameter-free OCO algorithms, then the resulting algorithm performs almost as well as the best one among the two. That is, (at least in the convex setting) we can select the optimal discount factor for free.
>
> We note that these three components crucially work together to guarantee good regret bounds in the convex setting. Since this is often the minimum requirement for any reasonable optimizer, we kept all three components in the nonconvex setting as well (otherwise the algorithm wouldn't make sense). This (ruling out design altenatives) is in our opinion a big benefit of theory-driven research.
>
> ## Minor issues
>
> - **On the convexity.** We essentially mean that although parameter-free OCO builds on the structure of convexity, it's intriguing to see that it works well on lifelong RL, a nonconvex and nonstationary task. We'll make this clearer.
> - **Stability.** Here we mean the iterates shouldn't drift too far. This is the same reason for the importance of weight decay in DL, which is somewhat widely accepted by the community.
>
>
> ## Questions
> - A round refers to an iteration.
> - We agree with you that the concept of discounted regret isn't so relevant to our exposition, so we'll remove it.
> - Loss rescaling means on Line 5 of Alg 2 (the tuner), we multiply $\beta$ during the update of $v_t$ and $\sigma_t$. The theory behind this is justified in (Zhang et al, 2024a).
> - **Distribution shift in Figure 7.** In the control environments, a distribution shift refers to altering the observation's dimensions by adding or subtracting random noise for a duration of 200 steps. This noise is changed every 200 steps. On the x-axis of Figure 7, each distribution shift number corresponds to a different 200-step interval. For example, Distribution Shift 1 covers steps 0-200, Shift 2 covers 200-400, and so forth.
> - **Cumulative reward sum.** The discounted regret is a suitable metric for the theoretical analysis in OCO, but in reality (experiments and real world), we only care about cumulative reward sum (as in almost all existing works). Somewhat inevitably, this is one of the common aspects where theory and practice differ.
> - **Uncertainty set $\mathcal{U}$.** Intuitively (in the context of lifelong RL) this could be thought as the loss landscape: since we only have gradient feedback, the loss landscape remains uncertain for our algorithm throughout its operation.
>
> Since parameter-free OCO is itself a quite sophisticated field, our exposition focused mostly on the high level main idea. Please let us know if you have any other concerns / comments, and we'll be glad to address those!

---

> > ### Comment · Reviewer_Jqmp · 2024-08-11
> > **Thank you for your response**
> >
> > I thank the authors for their explanations and answers. The authors agree with me that writing needs to be improved. I noticed the authors did not respond to my second concern, which an ablation study would have addressed. Thus, based on the current state of the paper, I have maintained my original score since I think major restructuring is needed, which will require another review.

---

> ### Author Response · Authors · 2024-08-12
> **Ablation Study Experiment and Writing Revisions**
>
> We thank you for your continued engagement with our work!
>
> # Ablation study
> Regarding your concern about the ablation study, as previously noted in our rebuttal, the components of our parameter-free algorithm are designed to function integrally, which is foundational to maintaining its parameter-free nature.
>
> However, we appreciate your suggestion that exploring the individual impact of each component could highlight their respective contributions.
>
> **We conducted an ablation study to quantify the impact of removing specific components, expressed as average normalized rewards over 25 seeds:**
> | Environment   | Full Configuration (Additive Aggregation & Erfi Tuner) | No Additive Aggregation | No Erfi tuner (1D Online Gradient Descent) | No Erfi Tuner or Additive Aggregation |
> |---------------|--------------------------------------------------------|-----------------|---------------------------|--------------------------------------|
> | **CartPole**  | 0.32                                                   | 0.19            | 0.02                      | 0.01                                 |
> | **Acrobot**   | 0.72                                                   | 0.45            | 0.00                      | 0.00                                 |
> | **LunarLander** | 0.54                                                | 0.49            | 0.16                      | 0.14                                 |
>
> We ablate the erfi tuner by replacing it with a baseline 1D online gradient descent tuner, where the parameter is $s_t$.  Additionally, we ablate the additive aggregation component by employing a single tuner with $\beta$=1.
>
> The ablation study demonstrates that the erfi tuner plays the most critical role in performance. Moreover, removing discounting diminishes the benefits of additive aggregation (which automatically selects the best discounting), though some advantages are retained in performance.
>
> ---
> # A simple revision for the clarity of writing concern
> We are pleased that our detailed explanation in the rebuttal has addressed your inquiries about how PACE works. We recognize that your concern is regarding the clarity of the writing in the method section. **To address this concern, we revise a part of Section 3.** Here is a modified excerpt we will include in the camera-ready text following your suggestions:
> > **PACE for Lifelong RL:** In lifelong RL, a key issue is the excessive drifting of weights $\theta_t$, which can detrimentally affect adapting to new tasks. To address this, PACE enforces proximity to a well-chosen reference point $\theta_{\mathrm{ref}}$, providing a principled solution derived from a decade of research in parameter-free Online Convex Optimization (OCO). Unlike traditional methods such as $L_2$ regularization or resetting, PACE avoids hyperparameter tuning, utilizing the unique properties of OCO to maintain weight stability and manage drift effectively.
> >
> > The core of PACE, similar to other parameter-free optimizers, incorporates three techniques:
> > 1. **Direction-Magnitude Decomposition**: Inspired by Cutkosky & Orabona (2018), this technique employs a carefully designed one-dimensional algorithm, the "parameter-free tuner," atop a base optimizer. This setup acts as a data-dependent regularizer, controlling the extent to which the iterates deviate from their initialization, thereby minimizing loss of plasticity, which is crucial given the high plasticity at the origin (Abbas et al., 2023).
> > 2. **Erfi Potential Function**: Building on the previous concept, the tuner utilizes the Erfi potential function, as developed by Zhang et al. (2024b). This function is crafted to effectively balance the distance of the iterates from both the origin and the empirical optimum. It manages the update magnitude by focusing on the gradient projection along the direction $\theta_t - \theta_{\mathrm{ref}}$.
> > 3. **Additive Aggregation**: The tuner above necessitates discounting. Thus, we employ Additive Aggregation by Cutkosky (2019). This approach enables the combination of multiple parameter-free OCO algorithms, each with different discount factors, to approximate the performance of the best-performing algorithm. Importantly, it facilitates the automatic selection of the optimal discount factor during training.
> >
> > These three components crucially work together to guarantee good regret bounds in the convex setting and are the minimum requirement for any reasonable parameter-free optimizer.
> >
> These revisions follow your suggestions on introducing the concepts utilized in the algorithm and explaining the interactions among the 3 components.
>
> We are happy to include the ablation study and writing revisions in the camera-ready version. **Do you have any other concerns or suggestions that were not addressed here?**

---

> > ### Comment · Reviewer_Jqmp · 2024-08-12
> > **Thank you for the ablation study**
> >
> > I thank the authors for providing the ablation study. This addresses my concern about the importance of the three components of PACE, and therefore, I raised my score to 6.

---

> > > ### Author Response · Authors · 2024-08-12
> > > **Thank you**
> > >
> > > Thank you so much for your thoughtful review and increasing your assessment!

---

### Official Review · Reviewer_346D · 2024-07-15

**Soundness:** 2
**Presentation:** 2
**Contribution:** 2
**Rating:** 6
**Confidence:** 3

**Summary:**

This work proposes a new method designed to mitigate the loss of plasticity when a model is continuously retrained on a set of tasks. The approach uses online convex optimization (OCO) via the minimisation of a regret function. The approach is compared against two baselines: PPO with ADAM (fixed learning rate) and CReLU. The benchmarks used are ProcGen, the Arcade Learning Environment (ALE) and Gym Control. The empirical tests show promising results with respect to standard reward metrics. Overall, the paper engages with a new and emerging field in machine learning that assume changing input distributions, multiple tasks in various contexts, e.g. supervised and reinforcement learning. Moreover, the paper addresses a recently discovered property of optimizers that reduce the ability of the network to continuously adapt to new tasks if continuously retrained (plasticity loss).

**Strengths:**

- The development of a parameter-free optimiser to enhanced performance under non-stationary condition is a interesting research venue in the scope of lifelong reinforcement learning.
- Addressing the issue of loss of plasticity in dynamic scenarios with shifting distribution is an emerging area in lifelong learning.
- The paper has a good set of benchmarks that are specific to lifelong reinforcement learning challenges.

**Weaknesses:**

1. The core of the novel algorithm is explained with pseudo code in Algorithm 1 and 2. I understand that the authors are very familiar with all the details of several papers by Cutkosky (at least 4 main sources), on which this work builds on
- direction-magnitude decomposition from (Cutkosky & Orabona, 2018),
- the additive aggregation from(Cutkosky, 2019),
- and the erfi potential function from (Zhang et al., 2024b).
-  (Cutkosky et al., 2023) for the recommended default parameter setup.

However, it's not obvious to me how clear the methodology is given the explanation in Section 3 to machine learning experts that are not very familiar with Cutkosky's work.

2. A clear highlight of the core technical novelty is missing in relation to previous work. I understand that this work "builds on" the cited work above, but it is generally not a sufficient statement of novelty. More effort is required to clearly spell the novel aspects with respect to previous work.

3. The most notable limitation that mostly affects my final evaluation is that this method for lifelong reinforcement learning (LRL) **is compared to only two baselines that are not lifelong reinforcement learning algorithms**. I understand that such choice might derive from the narrow focus of preventing plasticity loss. However, a strong paper in LRL needs to engage with the field more thoroughly, describe notable existing LRL, and how the proposed methods contributes to advancing the field. Candidate baselines could include RL implementations of EWC, Experience Relay based RL methods (e.g. Rolnick's CLEAR), PackNet by Mallya and Lazebink or Modulating Masks for LRL by Ben-Iwhiwhu, just to mentioned a few. I find it odd that in the limitation section, the suggestion is the future application of the algorithm to SAC and DQN, which again are not LRL algorithms.

4. As a follow-up to the previous limitation, the paper would benefit from engaging with the LRL field from the start, not only by describing the Spot example (which sounds like the authors are the first to highlight this to the community), but also describing methods that have been developed to address that problem over the last decades, their achievements, their limitations, and why this particular new approach is needed.

5. Limited analysis is provided on the inner-working of the algorithm (e.g. no ablation studies).

6. Improving the clarity of the hypotheses would benefit the paper. Is the hypothesis the fact that OCO can be used to reduce plasticity loss? Or that it can be used to reduce plasticity loss better than existing algorithms that do that? Or that it can be used to implement LRL? These questions are only partially answered or addressed in the paper.

**Questions:**

My questions are a direct follow-up on the limitations
1. Would you consider comparing your approach to lifelong reinforcement learning algorithms?
2. Why are all the three techniques from the parameter-free OCO required? What do they accomplish individually? Could you run ablation studies?
3. How does the ADAM default (adaptive learning rate) perform? I understand the need to freeze the learning rate to allow the proposed algorithm perform the adaptation, but it would be nevertheless valuable to assess the performance of the standard version of ADAM.

**Limitations:**

The limitation section is mainly focused on a narrow aspects, e.g. the need for a warm-up. However, this is also in contradiction with the Appendix in which it is suggested that such a warm-up is not required.

My conclusion is that the narrow scope of this limitations section stems from the lack of integration of this work with the broader field of lifelong reinforcement learning. Addressing the weaknesses that I pointed out, and engaging with a more thorough comparison of this approach with existing LRL approaches, a more insightful view of the pro and cons should emerge.

---

> ### Author Rebuttal · Authors · 2024-08-07
>
> Thank you for your insightful review and constructive feedback!
>
> ## Lifelong RL baselines
>
> Firstly, we acknowledge your comment on the lack of baselines. We recognize the important contributions of Ben-Iwhiwhu et al., 2023, Rolnick et al., 2019, Schwarz et al., 2018, and others in the field of lifelong RL. However, we didn't include these baselines since we consider our setting and scope to be a bit different (i.e., no task recycling, shared curricula, or privileged task boundaries; more on this soon). However, following your suggestion, we incorporated these baselines and found that the results (detailed below) support our hypothesis.
>
> ### Results
>
> Using open-source code from Ben-Iwhiwhu et al., 2023, we evaluate Adam and PACE across Impala, Online EWC, CLEAR, and Modulating Masks in our Procgen games. We modify the continual RL setup from Ben-Iwhiwhu et al. (2023) and Powers et al. (2022) to our lifelong RL setting (modifications are detailed in the Author Rebuttal).
>
> We present a comparison of average normalized rewards over 5 seeds for each method and game in Supplementary PDF Figures 1b-e and plot the average reward curves, excluding STD fill for clarity, in Supplementary PDF Figure 1a for Starpilot.
>
> In the Procgen environments, **PACE consistently outperforms Adam across various learning algorithms**:
>
> * IMPALA **by an average of 21.83%**
> * Online EWC **by an average of 15.86%**
> * CLEAR **by an average of 14.41%**
> * Modulating Masks **by an average of 10.14%**.
>
> In Supplementary PDF Figure 1a, we also observe when using Adam, *these methods are still vulnerable to loss of plasticity* in later levels (especially Online EWC).
>
> We are happy to include these results in our work and revise the introduction to acknowledge the history of these methods in the context of lifelong RL.
> ## Clarifying our Lifelong Setting
>
> * Our setting does not revisit old tasks nor assure any task similarity.
> * We operate in a strictly online setup, prioritizing the evaluation of rapid adaptation to new tasks without task boundaries (which mask-based and EWC-based continual RL approaches require).
> * In our setting, we demonstrate that plasticity loss occurs when adapting to new tasks never seen before.
>
>
> In contrast,
> * Ben-Iwhiwhu et al. (2023) focus on task recycling to balance learning new tasks with performance on previously seen ones. We consider this as a continual RL setting; indeed, Ben-Iwhiwhu et al., 2023 use "CORA: A Continual Reinforcement Learning Benchmark" (Powers et al., 2022) for Procgen experiments, where previously seen tasks are evaluated (backward transfer) during training.
> * In this same setting, Abbas et al. (2023) observe loss of plasticity *when tasks are revisited*.
>
>
> ## Clarifying our scope and hypothesis
> We would like to clarify our work specifically addresses the loss of plasticity problem in lifelong RL through the *lens of optimizers*, such as Adam.
>
> **Hypothesis**: Parameter-free optimization mitigates loss of plasticity more effectively than Adam, regardless of the learning algorithm or architecture employed (e.g., PPO, PPO with CReLU).
>
> **Conclusion:** We introduce a new parameter-free optimizer, PACE, and show it better preserves plasticity compared to Adam across 4 Procgen, 2 Atari, and 3 Gym Control lifelong settings.
>
> **We would like to note that PACE is orthogonal to learning or policy algorithms and architectures  (eg. PPO, CLEAR, Modulating Masks).** As seen in the results above, PACE can be integrated with other RL algorithms simply by replacing the optimizer (like Adam or RMSPROP) that they use to train with.
>
> **Evaluation Choice:** We opted to assess many environments with different input dimensions and distribution shifts, rather than many algorithms, to evaluate different modes of plasticity loss and isolate the effects of Adam and PACE. However, we have addressed your feedback on incorporating more baselines in Supplementary PDF Figure 1 and above.
>
> ## Comparison to parameter-free OCO
>
> We agree that our explanation of parameter-free OCO is somewhat abstract, given the space constraints. We will enhance more clarity in the camera-ready version!
>
> **On the (un-)necessity of an ablation study:** Parameter-free OCO has evolved over the last decade into a standard framework, primarily based on extending 1D algorithms to higher dimensions (Cutkosky & Orabona (2018)) and a data-dependent method for selecting the discount factor (Cutkosky (2019)). These contributions are critical for ensuring optimal performance in convex settings—considered the benchmark for optimizers—and are equally vital in nonconvex applications. Omitting these would undermine the algorithm’s effectiveness and theoretical motivations.
>
>
> **Our novelty.** When moving beyond convexity, existing works on parameter-free OCO consider training DL models. Intuitively, this doesn't fully utilize the capability of such algorithms: by design, OCO algorithms are able to handle even adversarially varying nature, whereas DL training is "stationary." Our work goes beyond this limitation by considering lifelong RL as an intrinsically "nonstationary" problem. This is a nontrivial "out-of-the-box" move for OCO, as the latter cannot capture the loss of plasticity we aim to address. In our opinion, the insight is pleasingly neat:
> - parameter-free OCO is all about better regularization and
> - better regularization mitigates loss of plasticity.
>
> Furthermore, PACE employs the 1D algorithm from Zhang et al. (2024), which is theoretically optimal yet untested in real-world deep learning training.
>
>
> ## Minor: Adam
>
> We use default PyTorch Adam, which we note is adaptive but *still* relies on an initial LR as a hyperparameter (in contrast, PACE is insensitive to LR).
>
> Please let us know if you have any other concerns/comments!

---

> ### Comment · Reviewer_346D · 2024-08-08
>
> The authors made a significant effort to improve the paper, in particular with the addition of baselines. Some concerns still remain, but the modifications grant an increase of my initial assessment.
>
> PS: I'm not able to access the new version of the paper. I'm not sure whether the authors have uploaded it.

---

> > ### Author Response · Authors · 2024-08-11
> >
> > Thank you for the thoughtful review and increasing your assessment! At the moment, we are not allowed to edit the original submission pdf, but we will carefully revise the camera-ready version to include an overview and results of these additional baseline methods.

---

### Official Review · Reviewer_eCfR · 2024-07-16

**Soundness:** 3
**Presentation:** 4
**Contribution:** 4
**Rating:** 7
**Confidence:** 4

**Summary:**

The paper tackles the problem of plasticity loss in RL, where a neural network may have reduced performance on learning new tasks after a task switch. The authors develop a hyperparameter-free algorithm, PACE, using ideas from online convex optimization. It can be interpreted as an automatically-tuned weight decay adapted during training. Experiments on large-scale benchmark tasks demonstrate the effectiveness of the approach compared to different baselines.

**Strengths:**

This paper builds a bridge between the fields of lifelong RL and online convex optimization (OCO), successfully using ideas from OCO to develop an algorithm addressing plasticity loss. It can be difficult to use fruitfully utilize theoretical ideas in a practical context so I commend the authors for that.

The paper is written clearly and generally easy to follow. Intuitions for the theory are provided which are helpful to understand the algorithm.
The evaluation is extensive and covers a wide variety of RL tasks, with larger-scale benchmarks such as ProcGen.
I also appreciate some of the more in-depth analyses such as inspecting the scaling $S_t$ over time in appendix C.

**Weaknesses:**

Generally, I have not identified any major weaknesses for this paper.
Some areas that could be improved (but are still acceptable):
- Only a couple of baselines are run, including weight decay and using CReLU activation.
	A suggestion would be to try layer norm since it is also parameter-free and has shown some benefits to address plasticity loss [1].
- While the algorithm is theoretically-motivated, I did not see any theoretical results included in the paper. If possible, it could be nice to have a theoretical guarantee for PACE even if it is in a different setting such as OCO. Such a result could give additional confidence that the algorithm is sound.

**Questions:**

_Clarification questions_
- Fig. 2: I do not fully understand what this figure is trying to convey. In particular, how is the figure demonstrating that PACE may be beneficial in this scenario? Maybe an explanation in the caption could help.

- For the section on tuning $\ell_2$ regularization (line 280), could you include a hyperparameter sensitvity plot i.e. performance vs. hyperparameter value? I would be curious to see how the performance varies around the best values.

- Fig. 6 In the experiments, it seems like even PACE suffers from some amount of plasticity loss since the learning curve does not reach the same maximum as in the first task (more evident in CartPole or LunarLander). Do you have any hypotheses to explain this?

_Comments and suggestions_


- Looking at the values of $S_t$ over time (in appendix C) they are usually around 0.01-0.1, which seems fairly small and have a distinct trend of decreasing over time.
This looks inversely correlated to the parameter norm, a quantity that is often measured in the context of plasticity loss. Have you looked into this relationship? I wonder if plotting $S_t \times ||\theta^{Base}||$ shows a roughly constant effective parameter size.

- If I understand PACE correctly, it looks like overall, it produces the next parameter $\theta_t$ through a weight $S_t$ which is used to obtain convex combination between the initial parameters and the base-optimized parameters $\theta^{Base}_t $.
What I find counterintuitive is that the gradients for the next step are then evaluated at $\theta_t$ but used to update $\theta^{Base}_t$, which could be quite different in terms of the policy implied.
Do you have any thoughts about this?

- A potential explanation, as discussed in [3], is that rescaling may also preserve the argmax for classification tasks (and the policy in this case) although it does affect the size of the logits. This would not explain effects on the value network though.

- On a related note, do the neural networks feature normalization layers e.g. layer norm?
When normalization layers are used, there is a scale-invariance introduced in the weights (well, if there are no bias units, see page 1. of [2]). I could see PACE being more effective in conjunction with normalization layers due to this rescaling operation not impacting (or having a small impact on) the function learned.

- About "near convexity", line 298. While the algorithm components are developed for the OCO setting, do you think it is possible that, instead of the problem being "near convex", it is the algorithms that have wider applicability than previous appreciated? That is, they may work well in a more general class of problems?

[1] "Understanding plasticity in neural networks" Lyle et al.

[2] "Understanding the Generalization Benefit of Normalization Layers: Sharpness Reduction" Lyu et al.

[3] "On Warm-Starting Neural Network Training" Ash and Adams

**Limitations:**

These are discussed.

---

> ### Author Rebuttal · Authors · 2024-08-07
>
> Thank you for your thoughtful review and for highlighting the strengths of our work!
>
> ## Layer normalization helps:
> Thank you for suggesting the use of layer normalization, as highlighted by Lyle et al. for addressing plasticity loss, in our experiments. We implement layer normalization in our control experiments for both PPO with Adam and PPO with PACE (Figure 3 in Supplementary PDF). Our results show that while layer normalization improves the performance of baseline Adam, it does not outperform PACE. Moreover, we find your hypothesis to be true: PACE with layer normalization performs better than PACE without layer normalization  (see Figure 3b in Supplementary PDF).
>
>
> ## Comments
>
> * Thank you for your close reading of our method and the appendix. We unfortunately did not have the opportunity to examine the relationship between the parameter norm and our scaling approach, but we are excited to explore this analysis in future work!
>
>
> * Regarding your question about updating $\theta_{t}^{Base}$ by evaluating the gradients at $\theta_{t}$, intuitively, there is more stability around the reference point. Therefore, updating the gradients from a base update near this safe region helps prevent issues with gradients that might otherwise explode or vanish if updated at $\theta_t$\.
>
> * We agree with your observation that the success of parameter-free optimization may not indicate a “hidden” convexity, but rather that these methods may have broader applicability. This is an important point, and future research should indeed investigate why these methods are effective in such settings.
>
> * Regarding Figure 2, we appreciate your feedback and will update the caption to better explain our visual abstraction of the setting.
>
> * You correctly pointed out that in Figure 6, while PACE combats plasticity loss, it may still be affected by it, as the reward does not reach the same level as during the first distribution shift. Although good, we believe PACE's scaling isn’t always optimal, especially in a non-convex setting. There may be a more "optimal" set of scaling values such that the performance is as good as the first task, and this is something that may evolve as parameter-free methods continue to improve.
>
>
> Please let us know if you have any other concerns or comments, and we'll be glad to address them!

---

> > ### Comment · Reviewer_eCfR · 2024-08-10
> >
> > After reading the rebuttal and other reviews, I remain satisfied with the paper and the additional experiments strengthen the paper further.
> > I am still recommending acceptance.

---

> > > ### Author Response · Authors · 2024-08-11
> > >
> > > Thank you for the insightful review and for reading our rebuttal response!

---

### Author Rebuttal · Authors · 2024-08-07

We thank all reviewers for their detailed feedback. We appreciate their recognition of our parameter-free approach to mitigating plasticity loss in lifelong RL as novel (Reviewers Jqmp, Jcag), its evaluation across a broad set of lifelong RL benchmarks (Jcag, 346D, eCfR), the extra analysis in the appendix (Jcag, eCfR), and the theoretical motivations behind our empirical idea (Jcag, eCfR). We have addressed specific questions and concerns in our individual responses and have included a **Supplementary PDF** with experiments aimed at addressing feedback and questions from reviewers 346D and Jcag. We include an overview of such experiments and their setups below.

## Overview of Figures and Experiments
### PACE mitigates plasticity loss on LRL baselines in Procgen (Figure 1 in Supplementary PDF) (for Reviewer 346D):

* **Figure 1a** demonstrates that while IMPALA Adam, Online-EWC Adam, CLEAR Adam, and Modulating Masks Adam exhibit loss of plasticity in later levels, IMPALA PACE, Online-EWC PACE, CLEAR PACE, and Modulating Masks PACE effectively mitigate this plasticity loss and enhance reward performance over subsequent distribution shifts. STD fill is omitted here for clarity but is included in the bar plots in **Figures 1b-e**.
* **Figures 1b-e** present average normalized rewards over 5 seeds across 120M timesteps for each method using PACE and Adam: **1b** for Starpilot, **1c** for Dodgeball, **1d** for Chaser, and **1e** for Fruitbot.

### PACE performs slightly better than Mechanic on all settings (Figure 2 in Supplementary PDF) (for Reviewer Jcag)
* We compare the average normalized reward for Mechanic and PACE for Procgen and Atari environments (over 10 seeds) and on Gym Control environments (over 25 seeds). We show here that Mechanic consistently underperforms PACE by a slight margin across all settings.


### Layer norm and Plasticity Injection can also help (Figure 3 in the Supplementary PDF) (for Reviewers Jcag, eCfR)

* **Figure 3a** demonstrates that both layer norm and plasticity injection effectively mitigate plasticity loss in CartPole with baseline Adam PPO.
* **Figure 3b** shows that in all three control environments, PACE outperforms both layer norm Adam and plasticity injection Adam. Additionally, combining layer norm with PACE further boosts the agent’s performance compared to using PACE without layer norm.


### Tuning weight decay can help, but not as much as PACE (Figure 4 in Supplementary PDF) (for Reviewer Jcag)
* **Figures 4a-c** plot the average normalized reward of different weight decays with Adam (over 15 seeds over 3000 timesteps) compared to PACE. The bar plots indicate that while different weight decay values with Adam can help, they consistently underperform in comparison to PACE for all 3 control environments.
* **Figures 4d** plots the best performing weight decay with Adam over 10 distribution changes for all 3 of the control environments. We observe that weight decay values are highly sensitive to the specific environment and distribution shift.

## Average Normalized Reward

We calculate the average normalized reward as depicted in the bar plots (**Figures 1b-e; Figure 2; Figure 3b; Figures 4a-c**) by following these steps:
* Normalize rewards based on the maximum and minimum values for each environment, irrespective of the method used.
* Compute the average reward for each timestep across all seeds.
* Average these values across all timesteps.
* *For Atari environments, normalization is also performed within each subtask due to varying reward scales across games.*



## Procgen Experimental Setup (for Reviewer 346D)

To align with our online lifelong RL setting described in the main paper’s Procgen experiments, we have made the following modifications from the continual RL setup used by Ben-Iwhiwhu et al., 2023, and Powers et al., 2022:

* We focus on individual games (e.g., Starpilot, Dodgeball, Chaser, Fruitbot) rather than cycling through multiple games for evaluation.
* Tasks are limited to a single level, preventing training and generalization across 200 levels.
* We have adjusted the Procgen distribution shift mode in the code from easy to hard.
* We switch the level every 8M timesteps.

---

### Decision · Program_Chairs · 2024-09-25

**Decision:**

Accept (poster)

**Comment:**

This paper introduces a parameter-free optimizer called PACE, using ideas from online convex optimization to target plasticity loss in lifelong RL. The novel approach is supported by strong theoretical motivation (although no theoretical guarantees) and extensive empirical evaluations across various environments. There were some initial concerns regarding clarity/presentation, the absence of certain baselines, and similar performance to some other methods (e.g., Mechanic), but these were mostly addressed in the rebuttal through the authors providing additional experiments, including an ablation study and more detailed statistical comparisons. There still seems to be only a slight improvement over Mechanic, but the reviewers in general agree that the paper’s contributions are valuable. The authors are advised to incorporate the reviewers’ suggestions to improve clarity/presentation, the new results, and issues raised in the discussion into the final version of their paper.